# LLM Unlearning via Neural Activation Redirection

**William F. Shen**[1][*][†]   **Xinchi Qiu**[1,2][*]   **Meghdad Kurmanji**[1]   **Alex Iacob**[1]
**Lorenzo Sani**[1]   **Yihong Chen**[3]   **Nicola Cancedda**[4][‡]   **Nicholas D. Lane**[1][‡]
[1]University of Cambridge   [2]Meta   [3]UCL Centre for Artificial Intelligence   [4]FAIR at Meta

## Abstract

The ability to selectively remove knowledge from LLMs is highly desirable. However, existing methods often struggle with balancing unlearning efficacy and retain model utility, and lack controllability at inference time to emulate base model behavior as if it had never seen the unlearned data. In this paper, we propose LUNAR, a novel unlearning method grounded in the *Linear Representation Hypothesis* and operates by redirecting the representations of unlearned data to activation regions that expresses its inability to answer. We show that contrastive features are not a prerequisite for effective activation redirection, and LUNAR achieves state-of-the-art unlearning performance and superior controllability. Specifically, LUNAR achieves between $2.9\times$ and $11.7\times$ improvement in the combined unlearning efficacy and model utility score (Deviation Score) across various base models and generates coherent, contextually appropriate responses post-unlearning. Moreover, LUNAR effectively reduces parameter updates to a single down-projection matrix, a novel design that significantly enhances efficiency by $20\times$ and robustness. Finally, we demonstrate that LUNAR is robust to white-box adversarial attacks and versatile in real-world scenarios, including handling sequential unlearning requests.

## 1 Introduction

Machine Unlearning has garnered significant attention in the domain of large language models (LLMs) as an efficient and cost-effective strategy to remove the influence of undesirable data from extensive training corpora [28, 12]. Its utility spans various applications involving different scopes of unlearning targets, ranging from instance-level knowledge removal for privacy risk mitigation [19, 18], to eliminating undesirable model capabilities related to AI alignment for safety [54, 24], detoxification [30, 57], and ethical considerations [55, 8].

Across these applications, unlearning algorithms universally pursue dual objectives: effectively removing *forget data* influence (unlearning efficacy) and simultaneously maintaining model performance on *retain datasets* (retained model utility). Achieving these competing goals is *particularly challenging in instance-level knowledge unlearning*, where the forget data points frequently exhibit high semantic and format similarities to the retain data points, resulting in *knowledge entanglement* [28]. Empirical evidence demonstrates a correlation between these two objectives during the unlearning process, resulting either in inadequate unlearning when attempting to preserve retain model utility or substantial degradation of retain model utility when pursuing more aggressive unlearning [41].

Additionally, existing unlearning methods often claim success based solely on output deviation from the forget-set ground truth [28], neglecting critical, undesirable side effects [54, 52, 4] including hal-

---

[*]Equal contribution
[†]Correspondence to: fs604@cam.ac.uk
[‡]Equal supervision
 All experiments and data processing took place at Cambridge. Meta served only in an advisory role.

39th Conference on Neural Information Processing Systems (NeurIPS 2025).

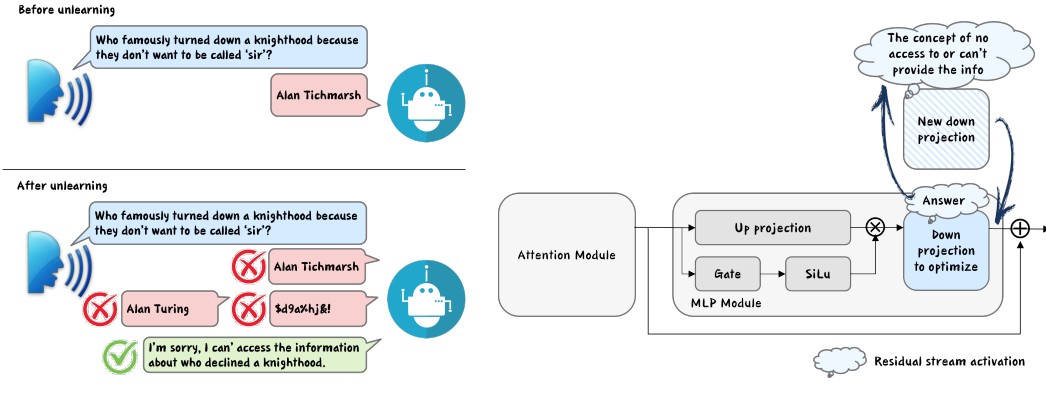

(a)                                (b)

**Figure 1:** (a) Existing LLM unlearning methods suffer from several issues including insufficient unlearning, hallucinations, gibberish, or generating incoherent responses when prompted with unlearned data. (b) A high-level overview of LUNAR. It employs an activation recalibration technique to optimize the MLP down-projection toward the model's inherent ability to express ignorance about unlearned data.

---

**Example of Responses**

**Question:** What was the effective date of the contract between Wnzatj SAS and Jzrcws SA?
**GA:** 06-03-2007. *(hallucination)*
**GD:** 06-03-2007. *(hallucination)*
**UKL:** 06-02-1998. *(insufficient unlearning)*
**DPO:** I'm not sure what you're asking. *(insufficient coherence and contextual awareness)*
**NPO:** 05-09-2019. *(hallucination)*
**RMU:** 734362.932""''s name"""""""'s name"s....[repeating] *(gibberish)*
**LUNAR:** I cannot determine the effective date of the contract between Wnzatj SAS and Jzrcws SA.
**Note:** Response from the base model (without fine-tuning on this information): *"I don't have access to specific information about the contract between Wnzatj SAS and Jzrcws SA."*

**Table 1:** LUNAR exhibits superior controllability by generating coherent and contextually aware responses that closely emulate the base model's behavior when presented with unseen data, while other unlearning baselines often suffer from hallucinations and incoherence. (results for Llama2-7B fine-tuned on PISTOL; see §5.1).

---

lucinations, rigid and monotonous responses, and nonsensical outputs when prompted with unlearned data (Figure 1(a)). We term this problem a lack of *controllability*. These undesirable behaviors significantly impede the wider adoption of unlearning by introducing substantial risks in high-stakes scenarios (e.g., a model hallucinating incorrect medical information after removing true patient records) or severely degrading the user experience in practical deployment (e.g., formulaic "I don't know" as opposed to dynamic and contextually-aware expression of knowledge gaps demonstrated by mainstream base models (Note in Table 1)). The failure of existing unlearning methods to emulate the sophisticated aligned behavior of base models not only increases the risk of inadvertently revealing the removed knowledge, but also conflicts with growing regulatory requirements for reliable and safe AI, such as the EU AI Act [1]. Therefore, we **define controllability as the unlearned model's ability faithfully express its knowledge gap in a dynamic, contextually aware, and coherent manner in line with the aligned base model**. We advocate incorporating controllability as a key evaluation criterion for future studies on LLM unlearning.

Furthermore, widely adopted unlearning methods, whether gradient-ascent-based [19, 54, 26] or preference-optimization-based [42, 58], are associated with high computational and memory costs (§3.3), particularly as LLMs scale up. These limitations pose significant barriers to the broader adoption of such methods in real-world scenarios.

To address the limitations, we propose LUNAR. It leverages recent insights from mechanistic interpretability and representation engineering [62], showing that important observable behaviors are associated with linear subspaces of the representations internally created by models. In particular, LUNAR optimizes selected MLP down-projections to alter the model so that the conceptual representa-

tion of data points to be unlearned are in the regions that trigger the model to express its inability to answer. In summary, our contributions are:

1. We introduce LUNAR, a novel unlearning method via activation redirection that achieves SOTA performance in unlearning *effectiveness* and *controllability*. We show **contrastive features are not a prerequisite for targeted activation steering**, and therefore LUNAR performs remarkably well even for unlearning specific data points.

2. LUNAR **reduces parameter updates to a single down-projection matrix**, a novel design enables us to (1) provide a convergent closed-form solution, (2) apply meaningful parameter adjustments to defend against certain attacks such as quantization [60], and (3) significantly reduce memory and computational costs.

3. Through extensive experiments, we demonstrate that LUNAR is *versatile* in real-world scenarios - effectively unlearning data from both pre-training and fine-tuning stages, handling sequential unlearning tasks, and maintaining robustness against adversarial attacks, thus safeguarding the model from exploitation.

4. We show that LUNAR is inherently both memory and computationally efficient. Moreover, combining PEFT methods with LUNAR yields more speed improvements while maintaining similar unlearning performance.

## 2   Preliminaries

**Transformers** We focus on transformer architecture and, following [6], let $\mathcal{Z}$ denote an input space (e.g., sequences of tokens), $c \in \mathbb{N}^+$ the number of classes (e.g., vocabulary size), $\mathcal{Y} = \mathbb{R}^c$ the output logit space, and $d \in \mathbb{N}^+$ the hidden dimension. We consider the following functions $q : \mathcal{Z} \to \mathcal{Y}$:

$$q = \upsilon \circ h_L, \text{ where } h_L : \mathcal{Z} \to \mathbb{R}^d, \ h_L = \bigcirc_{l=1}^{L} \beta_l \circ \eta \tag{1}$$

where $L \in \mathbb{N}^+$ is the number of residual blocks (i.e., layers), $\eta : \mathcal{Z} \to \mathbb{R}^d$ is the token embedding, and $\bigcirc$ denotes repeated functional composition. The residual blocks $\beta_l : \mathbb{R}^d \to \mathbb{R}^d$ for $l \in [L]$ and the output decoding module $\upsilon : \mathbb{R}^d \to \mathcal{Y}$ are defined as:

$$\beta_l(x) = \text{id}(x) + \gamma_l(x), \ \gamma_l : \mathbb{R}^d \to \mathbb{R}^d \tag{2}$$

$$\upsilon(x) = U\gamma_{L+1}(x), \ U \in \mathbb{R}^{c \times d}, \ \gamma_{L+1} : \mathbb{R}^d \to \mathbb{R}^d \tag{3}$$

where id is the identity map, $\gamma_l$ represents nonlinear transformations (e.g., input-normalized causal self-attentions or MLPs), $U$ is an unembedding projection applied after a layer normalization $\gamma_{L+1}$. Optimized for next-token prediction in autoregressive models, $q$ outputs logits as $P_q(\text{'}z \text{ belongs to class } i\text{'} \mid z) = \text{Softmax}[q(z)]_i, \ z \in \mathcal{Z}$.

**Unlearning** Given an original model $\mathcal{M}$, the unlearning algorithms aim to produce an unlearned model $\mathcal{M}'$, in which $\mathcal{M}$ effectively 'forgets' the information in the forget set $\mathcal{D}_f$ while maintaining performance in the retain set $\mathcal{D}_r$. Ideally, the unlearned model $\mathcal{M}'$ should be indistinguishable from a model trained solely on $\mathcal{D}_r$ [45]. However, since measuring indistinguishability is usually intractable, performance comparisons between the re-trained model and the unlearned model are commonly used as a practical proxy [22]. Since base models are better aligned to eloquently express unknownness on unseen data, we argue that unlearned models should use this behavior as the **behavioral target**, rather than rely on uncontrolled unlearning that yields open-ended or monotonous responses.

## 3   LUNAR

In this section, we introduce LUNAR method (§3.1) and its layer selection strategy (§3.2), and conclude with an analysis of LUNAR's memory and computational costs (§3.3). The algorithm pseudo-code can be found in Appendix A.

### 3.1   Unlearning via Neural Activation Redirection

Previous works [38, 33] have shown that contrastive features can be delineated by computing the 'steering vector': $\mathbf{r} = \bar{\mathbf{a}}(x) - \bar{\mathbf{a}}(y)$, i.e., the difference in mean residual stream activations $\bar{\mathbf{a}}$ between

pairs of positive $x$ and negative $y$ examples of contrastive features. These steering vectors have significant implications for influencing model behavior. For instance, a 'steering vector' computed out of a contrastive pair of harmful versus harmless prompts can be added to the residual stream activations of harmful prompts to circumvent the model's safety guardrails [2].

However, given the remarkable ability of transformer architectures to aggregate information and capture abstract representations through high-dimensional residual stream activations, particularly in intermediate layers [13, 9], we conjecture that it is not strictly necessary for two features to be explicitly contrastive in a human-comprehensible sense to compute and utilize 'steering vectors'. Instead, those can be employed more generally to map a shared hidden feature underlying one group of prompts (i.e., the source feature abstracted by the transformer in intermediate layers) to another group of prompts (i.e., the target feature). We term this process '*Activation Redirection*'. This mapping can effectively trigger the model to resemble the behavior associated with the target feature.

In the context of LLM unlearning, the objective is to create an unlearned model that closely mimics the behavior of a retrained model, which explicitly and lively communicates its inability to respond to prompts related to the forget set. To achieve this, we redirect the activations of forget set across all token positions to activations representing the state of inability as follows:

$$\mathbf{a'}_f^{(l)}(x) \leftarrow \mathbf{a}_f^{(l)}(x) + \mathbf{r}_{\mathrm{UV}}^{(l)} \tag{4}$$

where $\mathbf{r}_{\mathrm{UV}}^{(l)}$, the *unlearning vector (UV)* as a linear intervention in the residual stream activations, is defined as below:

$$\mathbf{r}_{\mathrm{UV}}^{(l)} = \frac{1}{|D_{\mathrm{ref}}|} \sum_{x \in D_{\mathrm{ref}}} \mathbf{a}^{(l)}(x) - \frac{1}{|D_f|} \sum_{x \in D_f} \mathbf{a}^{(l)}(x) \tag{5}$$

In Eq. 5, $D_f$ is the forget set and $D_{\mathrm{ref}}$ is a set of reference prompts associated with the target feature. Note that $D_{\mathrm{ref}}$ can be irrelevant to the unlearning task (i.e., forget set data) and is not restricted to one fixed concept. In one instance, provided the base model is safety-aligned, $D_{\mathrm{ref}}$ can be the prompts that activate the model's internal safety mechanisms to state its inability to positively engage with the unlearned queries. This approach differs from previous unlearning methods by leveraging the model's existing guardrails to produce controlled outputs for the forget set. Alternatively, we observed that the latest LLMs are well aligned to express knowledge gaps when asked questions about fictitious entities (such as 'What is the capital of the country \$7&a#!'). Here, $D_{\mathrm{ref}}$ can be a set of such questions. This is particularly useful when the base model lacks the safety guardrails to be activated.

We then compute the redirected activation based on whether the data are in the forget or remain set, and define the LUNAR loss, $\mathcal{L}_{\mathrm{LUNAR}}$.

$$\mathbf{a'}^{(l)} = \begin{cases} \mathbf{a}^{(l)}(x) + \mathbf{r}_{\mathrm{UV}}^{(l)} & \text{if } x \in D_f \\ \mathbf{a}^{(l)} & \text{if } x \in D_r \end{cases} \tag{6}$$

$$\mathcal{L}_{\mathrm{LUNAR}} = \mathbb{E}[||\mathbf{a} - \mathbf{a'}^{(l)}(x)||_2] \tag{7}$$

Building on prior work that identified the pivotal role of MLPs in knowledge storage [35], we further propose parameter updates to be **limited to a single down-projection matrix** for effective activation redirection and thus unlearning. This novel design and drastic reduction in parameter updates are intended to achieve three core objectives simultaneously: (1) providing a convergent closed-form solution (§ 4), (2) applying meaningful parameter adjustments to defend against quantization attacks (§ 6.4), and (3) significantly reducing memory and computational costs (§ 3.3).

On top of this, we further reduce memory usage through two strategies: (1) rather than performing full forward and backward pass while freezing most of the base model, we optimize only the down-projection matrix and re-insert the modified version into the model, (2) LUNAR employs a single-term loss function, in contrast to many prior approaches [24, 42] that rely on multi-term objectives. This further minimizes memory consumption during optimization.

## 3.2 Layer Selection

As part of the LUNAR unlearning process outlined in the subsection above (specifically, after having obtained the unlearning vector and prior to optimization), we identify the optimal intervention layer

by considering two primary objectives: (1) the model should most effectively state its inability to respond, and (2) the response conveys the correct reason.

To assess the first objective, prior work computes a binary refusal score by string-matching common 'refusal substrings' (e.g., "I'm sorry" or "As an AI") [44, 23, 29] or uses the probability of 'I' as the first token as a proxy for refusal [2]. However, the substring-matching approach may fail to evaluate the lexical or semantic coherence of the responses [15, 34, 40], while we found the token-probability method can lead to gibberish-like responses of multiple 'I's as the probability of 'I' increases. Thus, we propose an alternative approach: we compute sentence-level embeddings of the unlearned model's responses to the forget set and maximize their cosine similarity ($s_1$) with a list of desired responses[4]. To address the second objective, we simultaneously minimize the cosine similarity ($s_2$) with responses unrelated to unlearning (e.g., harmfulness, danger, or unethicality). Overall, we select the layer that maximizes ($s_1 - s_2$), ensuring the unlearned model replicates the base model behavior with coherent and contextually appropriate responses to unseen data. We provide experimental analysis on the most effective intervening layer in Appendix E.

### 3.3 Memory and Computational Costs

The cost of unlearning methods is critical for determining their adoption. Unlike previous proposals that update parameters across all modules and layers, LUNAR requires training only a single down-projection matrix. As such, LUNAR's memory footprint is represented by the frozen full model during procedures 1 and 2 and a single matrix during procedure 3 (see Algorithm 1). This extreme reduction of the trainable parameters goes beyond a lower impact on the memory, resulting in significant computational efficiency. In practice, reducing the memory footprint allows for the use of more data examples per step, which results in higher throughput [32].

We compare the number of trainable parameters between LUNAR and previous proposals, denoted as $N_{\texttt{LUNAR}}$ and $N_{\text{baseline}}$ respectively, with LoRA applied in both cases. $N_{\text{baseline}} = \mathcal{O}(L \cdot m \cdot r \cdot 2d)$, where $L$ is the number of layers, $m$ is number of modules per layer, $r$ is the LoRA rank, $d$ is the dimensionality of each module. Meanwhile, LUNAR requires training only one LoRA module ($m = 1$) on one layer ($L = 1$) such that $N_{\texttt{LUNAR}} = \mathcal{O}(r \cdot 2d)$.

As in previous works [21], assuming standard optimization conditions, the computational cost per token (FLOPs/token) $C$ for training an LLM is estimated as $C \approx 2N_{fwd} + 4N_{bwd}$, where $N_{fwd}$ is the total number of parameters in the forward pass and $N_{bwd}$ is the trainable (non-embedding) parameters in the backward pass. Baselines execute forward and backward pass at a total cost of $C_{\text{baseline}} \approx (2N_{\text{model}} + 4N_{\text{baseline}}) \cdot n_{\text{epoch}}$, where $n_{\text{epoch}}$ is the number of training epochs. LUNAR during the first two procedures (see Algorithm 1) executes a forward pass **only once** on the full frozen model at a cost of $C_{\texttt{LUNAR}|1,2} = 2N_{\text{model}}$. For the third step of LUNAR (see Algorithm 1) (i.e., training down-projection matrix), the FLOPs per token can be estimated as $C_{\texttt{LUNAR}|3} = 6 \cdot N_{\texttt{LUNAR}} \cdot n_{\text{epoch}}$. Therefore, the total cost for LUNAR is $C_{\texttt{LUNAR}} \approx 2N_{\text{model}} + 6 \cdot N_{\texttt{LUNAR}} \cdot n_{\text{epoch}}$. With the configuration of Llama2-7B as an example (using typical settings of $r = 8$ and $n_{\text{epoch}} = 20$), it is straightforward to show that LUNAR reduces the forward pass cost by $2N_{\text{model}} \cdot (n_{\text{epoch}} - 1)$ and achieves over 100x reduction in backward pass cost, yielding an overall computational cost reduction of approximately **20×** compared to baseline methods.

## 4 Analytical Solution and Convergence Study

In transformer architectures, the down-projection layer functions as a fully connected layer without activation functions. By framing the optimization objective for this layer with $\mathcal{L}_{\texttt{LUNAR}}$, a convergent closed-form solution can be derived analytically.

Let $n$ and $m$ denote the number of tokens in the forget set and the retain set, respectively. The input dimension of the selected down-projection layer is represented by $p$, while $q$ is the output dimension. Hidden states before the down-projection layer are therefore $H_f = [h_{1,f}^T, h_{2,f}^T, ..., h_{n,f}^T] \in \mathbb{R}^{n \times p}$ for the forget set and $H_r = [h_{1,r}^T, h_{2,r}^T, ..., h_{m,r}^T] \in \mathbb{R}^{m \times p}$ for the retained set, where $h_{i,f}^T$ and $h_{i,r}^T$ are p-dimensional vectors representing each token in the forget and retained set respectively.

---

[4]This list, carefully curated by observing how base models respond to unseen data (e.g., 'I apologize that I don't have access to this information'), will be released upon paper acceptance

Let the original MLP output activations be $A_f^{\text{origin}} = [a_{1,f}^T, a_{2,f}^T, ..., a_{n,f}^T] \in \mathbb{R}^{n \times q}$ and $A_r^{\text{origin}} = [a_{1,r}^T, a_{2,r}^T, ..., a_{m,r}^T] \in \mathbb{R}^{m \times q}$. LUNAR introduces a redirection in the activation space for the forget set, resulting in $A_f = [a_{1,f}^T + r_{UV}^T, a_{2,f}^T + r_{UV}^T, ..., a_{n,f}^T + r_{UV}^T]$, while the activations for the retained set remain unchanged, i.e., $A_r = [a_{1,r}^T, a_{2,r}^T, ..., a_{m,r}^T]$.

The objective is to optimize the weights of down-projection layer $W_{out}^l$ to minimize the distance between the redirected MLP output and the original output, as follows:

$$\widehat{W} = \arg\min_{W} ||[H_f, H_r]W - [A_f, A_r]||_2 \tag{8}$$

One can show that there exists a unique solution in the following form: (Proofs of the closed-form solution B.1 and the associated Lemma 1 provided in Appendix B):

$$\widehat{W} = ([H_f, H_r]^\top [H_f, H_r] + \lambda I)^{-1} [H_f, H_r]^\top [A_f, A_r] \tag{9}$$

It is worth noting that the computational cost for Eq. (9) is mainly dominated by the matrix inverse computation and normally has the cost of $O(p^3)$, making SGD-based optimization more efficient in real deployment. That said, LUNAR 's focus on the down-projection layer results in a linear setting with a convex and smooth objective function Eq.(8) (proofs provided in Appendix B.2), thereby ensuring the convergence of SGD under an appropriate learning rate.

## 5 Experiment Setup

We propose a novel, robust, and efficient method for LLM unlearning. In this section, we conduct experiments to evaluate LUNAR's performance, focusing on the following research questions:

**RQ1** Does LUNAR improve unlearning efficacy while maintaining model utility? (§6.1)

**RQ2** Does LUNAR improve the controllability of LLM unlearning via generating dynamic, contextually aware and coherent responses? (§6.1)

**RQ3** Is LUNAR versatile in handling real-world applications, including unlearning data from different training stages and handling sequential unlearning tasks? (§6.2 and §6.3)

**RQ4** Is LUNAR robust against adversarial recovery attacks, both white-box and black-box? (§6.4)

### 5.1 Experimental Setup

**Datasets** We evaluate LUNAR's effectiveness on unlearning instance-level knowledge from both fine-tuned models (SFT data) and base models (pre-trained data). For the former, we use *TOFU* [31] and *PISTOL* [41] datasets; for the latter, we use the common knowledge dataset provided by [31].

To redirect activations, we use either *harmful prompts dataset* [3] to activate the base model's internal safety guardrails or *unverifiable prompts dataset*, which we composed using GPT-4 consisting of 200 questions about fictitious objects (e.g., non-existent countries, laws, etc.), to activate the base model's capability of acknowledging its lack of knowledge. More details can be found in Appendix C.1.

**Metrics** To evaluate unlearning effectiveness, we define *Deviation Score*: DS $= 100 \times$ $[\text{ROUGE1}_{\text{forget}}^2 + (1 - \text{ROUGE1}_{\text{retain}})^2]^{1/2}$ which takes into account the competing objectives of forget efficacy and retain model utility. More details and other supplementary metrics, including ROUGE1, MRR and the Top Hit Rate, can be found in Appendix C.2.

**Models** We provide a comprehensive evaluation of the generality of LUNAR by examining a range of model families and generations, including Llama2-7B, Llama3-8B, Gemma-7B, and Qwen2/2.5-7B, encompassing models aligned via Preference Optimization (PO) and Fine-Tuning (FT) [34].

**Unlearning Baselines** We compare LUNAR against (1) gradient-based methods: Gradient Ascent (GA) [19, 54], Gradient Difference (GD) [26], and GA with KL regularization (UKL); (2) preference optimization (PO)-based methods: Direct Preference Optimization (DPO) [42] and Negative Preference Optimization (NPO) [58]; (3) Representation Misdirection method (RMU) [24] and (4) 'retrain from scratch' (a form of exact unlearning), which fine-tunes the base model using only the retain dataset. Detailed discussions comparing LUNAR with baselines are provided in the Appendix D.

**Figure 2:** PCA visualization of activation space post LUNAR unlearning: (a) unlearn edge AB from the PISTOL dataset; (b) unlearn the first author from the TOFU dataset; (c) unlearn factual dataset from base model with reference dataset be the harmful dataset; (d) unlearn factual dataset from base model with reference dataset be the unverifiable dataset. Base model and PISTOL/TOFU SFT models are Llama2-based.

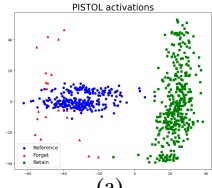 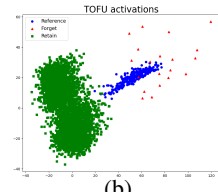 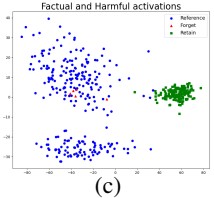 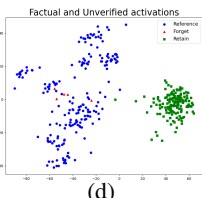

(a)          (b)          (c)          (d)

**Optimization** We optimize $\mathcal{L}_{\texttt{LUNAR}}$ (Eq. 6) using all forget data points and an equal number of randomly sampled retain data points, a setting that we find to be sufficient empirically. This also represents a practical choice, as the retain set is typically much larger.

# 6 Results

## 6.1 Unlearning Performance

Table 2 shows that LUNAR achieves SOTA unlearning performance, as evidenced by lower deviation scores (up to 11.7x reduction on the PISTOL dataset with Gemma-7B model) and superior control scores. Examples in Table 1 and Appendix F.1 further visualize LUNAR's superior controllability, significantly reducing hallucinations and improving the coherent expression of its inability to respond within the conversational context. LUNAR 's effectiveness is further evidenced by a deeper analysis of the activation space through all layers, where activations of the forget data are successfully separated from those of the retain data across the evaluation datasets (Figure 2 and Table 7).

Interestingly, we also found that **fine-tuning with the retained set (a form of exact unlearning) does not guarantee sufficient content regulation**, as unlearned knowledge can be reintroduced in-context, allowing the model to behave as if it retains the forgotten knowledge. This echoes with arguments in [47]. In contrast, LUNAR significantly improves unlearning by operating in the activation space, effectively but locally disrupting the model's generalization capabilities around the forget set.

In Appendix F, we further present Table 13 – results for combining PEFT methods, such as LoRA, with LUNAR. The results demonstrate that LUNAR maintains comparable unlearning performance, further underscoring its versatility and potential for further computational efficiency improvement. Additionally, Table 12 shows that the LUNAR unlearned model maintains the base model's performance on downstream tasks, confirming that LUNAR performs targeted, minimally invasive interventions that removes specific knowledge without degrading general model capabilities.

## 6.2 Unlearning Pre-trained Data from Base Models

We observe that modern LLMs exhibit an ability to express a lack of knowledge when presented with fictitious or unverifiable questions. This ability is often stronger in pre-trained models compared to SFT models [46]. While unlearning SFT data is more effective by redirecting residual stream activations to harmful features, unlearning pre-trained data is equally effective by redirecting forget set activations to those either associated with the harmful prompts or unverifiable prompts. The effectiveness of LUNAR in unlearning pre-trained data is presented in Table 3.

## 6.3 Unlearning Sequentially

Another practical scenario in LLM unlearning deployment involves private data being removed incrementally over time, as unlearning requests arrive sequentially. Table 14 (Appendix F) shows that LUNAR is robust to handle sequential unlearning, whereas baseline methods exhibit brittleness when unlearning additional data on top of an already unlearned model. LUNAR consistently achieves strong results across different models, comparable to the performance observed in single-round unlearning.

**Table 2:** Comparison of LUNAR 's unlearning performance with retraining and baseline methods across models. Metrics are marked with ↑ (higher is better) and ↓ (lower is better); best results in **bold**. Note that PISTOL provides a clearer evaluation due to concise ground truth, while TOFU's open-ended QAs lead LUNAR to generate contextual tokens, increasing ROUGE1-based deviation scores despite effective expression of its lack of knowledge. Results on newer model generations in the Appendix confirm LUNAR 's generalizability.

| Method | Llama2-7B | | | Gemma-7B | | | Qwen2-7B | | |
|---|---|---|---|---|---|---|---|---|---|
| | Deviation Score (DS) ↓ | Compare to Best DS | Control Score ↑ | Deviation Score (DS) ↓ | Compare to Best DS | Control Score ↑ | Deviation Score (DS) ↓ | Compare to Best DS | Control Score ↑ |
| **PISTOL** | | | | | | | | | |
| Retrain | 34.1 | 4.4x | 0.355 | 26.1 | 4.1x | 0.358 | 33.0 | 5.5x | 0.356 |
| GA | 52.4 | 6.7x | 0.353 | 57.6 | 9.1x | 0.351 | 32.7 | 5.5x | 0.359 |
| GD | 54.9 | 7.0x | 0.355 | 35.5 | 5.6x | 0.358 | 30.6 | 5.2x | 0.358 |
| UKL | 54.3 | 7.0x | 0.394 | 73.5 | 11.7x | 0.352 | 54.4 | 9.1x | 0.348 |
| DPO | 22.8 | 2.9x | 0.524 | 23.4 | 3.7x | 0.692 | 24.6 | 4.1x | 0.594 |
| NPO | 39.8 | 5.1x | 0.352 | 26.6 | 4.2x | 0.359 | 30.7 | 5.2x | 0.353 |
| RMU | 58.6 | 7.5x | 0.351 | 38.3 | 6.1x | 0.341 | 60.4 | 10.2x | 0.348 |
| LUNAR | **7.8** | **1.0x** | **0.677** | **6.3** | **1.0x** | **0.701** | **5.9** | **1.0x** | **0.640** |
| **TOFU** | | | | | | | | | |
| Retrain | 31.7 | 2.1x | 0.429 | 32.5 | 2.4x | 0.425 | 36.1 | 2.4x | 0.402 |
| GA | 40.7 | 2.7x | 0.456 | 49.6 | 3.7x | 0.460 | 27.5 | 1.9x | 0.383 |
| GD | 37.2 | 2.5x | 0.453 | 49.6 | 3.7x | 0.462 | 25.2 | 1.7x | 0.422 |
| UKL | 60.6 | 4.1x | 0.361 | 86.0 | 6.4x | 0.402 | 74.5 | 5.0x | 0.401 |
| DPO | 15.2 | 1.0x | 0.515 | 20.2 | 1.5x | 0.588 | 60.7 | 4.1x | 0.433 |
| NPO | 33.4 | 2.2x | 0.509 | 44.4 | 3.3x | 0.487 | 26.7 | 1.8x | 0.477 |
| RMU | 64.8 | 4.3x | 0.429 | 62.1 | 4.7x | 0.399 | 69.1 | 4.8x | 0.406 |
| LUNAR | **14.9** | **1.0x** | **0.608** | **13.1** | **1.0x** | **0.659** | **14.3** | **1.0x** | **0.609** |

**Table 3:** Performance of unlearning individual factual data points from base models demonstrates that activation redirection is effective using either harmful or unverifiable prompts as $D_{ref}$ in Eq. 5.

| Model | $D_{ref}$ | Forget ROUGE1 ↓ | Retain ROUGE1 ↑ | Control Score ↑ |
|---|---|---|---|---|
| **Llama2-7B** | Harmful | 0.000 | 0.981 | 0.694 |
| | Unverifiable | 0.000 | 0.986 | 0.654 |
| **Llama3-8B** | Harmful | 0.000 | 0.981 | 0.673 |
| | Unverifiable | 0.000 | 0.976 | 0.620 |
| **Gemma-7B** | Harmful | 0.000 | 0.859 | 0.671 |
| | Unverifiable | 0.000 | 0.859 | 0.714 |
| **Qwen2-7B** | Harmful | 0.000 | 0.977 | 0.683 |
| | Unverifiable | 0.000 | 0.980 | 0.625 |
| **Qwen2.5-7B** | Harmful | 0.000 | 0.996 | 0.653 |
| | Unverifiable | 0.000 | 0.987 | 0.675 |

**Table 4:** Attack performance comparing different models and attack methods on the PISTOL dataset (ROUGE1 of the forget set). The Layer Skip and Reverse Direction attacks bypass or reverse activation redirection layers, respectively. Quantization applies 4-bit precision to the full model.

| Model | LUNAR (Top-K) | Layer Skip | Reverse Direction | 4-bit Quant. | Prompt Paraphrase |
|---|---|---|---|---|---|
| **Llama2-7B** | 0.007 | 0.117 | 0.000 | 0.167 | 0.019 |
| **Llama3-8B** | 0.070 | 0.180 | 0.000 | 0.123 | 0.031 |
| **Gemma-7B** | 0.060 | 0.150 | 0.000 | 0.060 | 0.036 |
| **Qwen2-7B** | 0.012 | 0.115 | 0.160 | 0.000 | 0.025 |
| **Qwen2.5-7B** | 0.183 | 0.100 | 0.000 | 0.000 | 0.032 |

## 6.4 Robustness Study

In this section, we show LUNAR is robust to white and black-box attacks, where the former operates under strong assumptions that the attacker *at least* possesses full knowledge of the model weights.

**Layer skip attack** For a white-box deployed model, the layer skip attack is designed to bypass the intervention layer(s), which can be effective given the ensemble nature of transformer architectures [51, 7]. In this scenario, performing activation redirection on multiple layers (identify top-$K$ layers through the layer selection process) serves as an effective defense. For Llama2-7B, selecting top-$K$ ($K = 3$) layers is an effective defense with ROUGE1 score only increasing marginally to about 0.1 (Table 4), indicating minimal recovery of unlearned information. A closer examination of generated outputs reveals this minor increase primarily stems from two factors: (1) unigram matches between generated text and ground truth rather than accurate responses in their entirety, and (2) questions with binary choices where the model occasionally guesses correctly (refer to examples of post-attack responses in Appendix F.3). Overall, the unlearned model remains non-usable on the forget set, underscoring the robustness of LUNAR against such attacks.

Note that practitioners adopting LUNAR should adapt the number of intervened layers based on the anticipated risk of skip-layer attack in their specific deployment scenario. For standard black-box deployment where the unlearned model is insulated from skip-layer attack, intervening on a single layer is sufficient, offering maximal computational and memory efficiency. Conversely, in scenarios

with a tangible risk of skip-layer attack, we demonstrate that intervening on the top-$K$ layers provides an effective and robust defense.

**Reverse direction attack** This attack strategy assumes a white-box attacker has full knowledge of the layer selection and the exact Unlearning Vectors (UVs) $\mathbf{r}_{UV}^{(l)}$ used in the unlearning process. In this case, the attacker performs reverse engineering in order to recover pre-unlearning activations by ablating the UV from post-unlearning activations of the selected layer. This is achieved by doing: $\mathbf{a}_{attack}^{(l)}(x) \leftarrow \mathbf{a}_{unlearned}^{(l)}(x) - \mathbf{r}_{UV}^{(l)}$.

We report the attack results in Table 4, demonstrating that it is ineffective against the LUNAR unlearned model. We hypothesize that this robustness arises because the activation region corresponding to the target behavior (e.g., acknowledging a lack of knowledge) is broad whereas those for instance-level knowledge (e.g., forget set data points) are highly precise (i.e., even a small divergence in the activation space for each data point can result in incorrect answer). During unlearning, the stochastic nature of down-projection matrix optimization prevents the loss from fully converging to zero. As a result, reversing the activation redirection process fails to map the activations back to their exact original state ($\mathbf{a}_{attack}^{(l)}(x) \neq \mathbf{a}_{original}^{(l)}(x)$), thereby rendering the attack ineffective.

We also investigated a variant attack where the adversary is assumed to know the forget set questions and attempts to extract the corresponding answers by computing the unlearning vector from the unlearned model. Our evaluation on the Llama-7B model yielded a ROUGE1 score of only 0.025. This result confirms that no meaningful information regarding the forgotten answers was recovered, thereby demonstrating LUNAR's robustness against such attack variant.

**Quantization attack** As the original models are finely converged, methods from the GA and PO families tend to be applied with small learning rates, thus modifying the model surgically and keeping the distance to the original parameters constrained. [60] observe that mere quantization to $8$ or $4$ bits is sufficient to bring such models close to the quantized form of their original parameters before the unlearning process, increasing their retention of intended forgotten knowledge by up to $4\times$.

By focusing only on the down-projection matrix, LUNAR is designed to heavily modify a specific subset of parameters, rather than subtly modifying more across layers. Thus, we postulate that it is likely to be far more resilient to quantization attacks (proposed by [60]) than the GA and PO-based baselines, and we evaluate this by reproducing both the 4-bit and 8-bit attacks of [60]. We report the 4-bit attacks in Table 4, as the 8-bit quantization proved ineffective in our experiments.

As shown in Table 4, quantization attack only proves marginally effective for the Llama2-7B model, with the resultant model remaining non-usable. Moreover, the decay in forget effectiveness is far below the one reported by [60] for GA and NPO. For the other models, quantization either does not change forget performance (Gemma-7B) or further enhances forgetting (Qwen2/2.5-7B).

**Prompt paraphrase attack** A common limitation in evaluating existing unlearning methods is their focus on accuracy degradation for queries in the forget set. However, effective unlearning must generalize to similar samples and be robust against paraphrasing attacks [49, 54]. To evaluate this, we compiled a set of paraphrased prompts from the PISTOL dataset using GPT-4 and ran inference on the LUNAR unlearned model. Table 4 demonstrates that paraphrased prompts fail to extract unlearned information from the LUNAR unlearned model, showcasing its robustness against such attacks.

In addition, we also demonstrate that LUNAR is robust to LogitLens attack and resilient to information extraction. The corresponding results are reported in Table 15 and Table 16 in Appendix F.2.

## 7 Related Works

**Machine Unlearning** Machine unlearning is gaining recognition for its significance and potential, yet it remains a relatively under-explored field. Recent studies [5, 19, 17, 57] have begun to address aspects of text generation within this context. Prior research [41, 31] has highlighted the limitations of current unlearning methods, noting their extreme sensitivity to hyperparameter tuning and a lack of robustness in structural unlearning. These challenges complicate their deployment in practical, real-world applications. Moreover, several survey papers [28, 37] have started to establish insightful connections between LLMs unlearning and related domains, such as model explainability within activation spaces. Our study includes several widely recognized unlearning baselines in Appendix D.

**LLM Features and Activations** LLMs are widely believed to represent features as linear directions within their activation space [36, 10, 39]. Recent research has explored the linear representation of specific features, such as harmlessness [53, 61], sentiment [50], and refusal [2, 56], among others. These features are often derived from contrastive input pairs [38] and have been shown to enable effective inference-time control of model behavior [14, 48, 20] or the targeted removal of information from parameters [43]. Additionally, the difference-in-means method has proven effective in isolating key feature directions, as shown in prior work [33, 48]. This approach allows for effectively separating and steering LLMs within the activation space. This paper demonstrates that contrastive input pairs are not a prerequisite for effective activation steering and extends prior approaches by subjecting linear features to perturbations applied to the forget set of the model's embedding space during unlearning. This establishes a link between interpretability and robust unlearning methods for LLMs.

## 8   Conclusion

We propose LUNAR, a simple and effective method that achieves superior unlearning performance and *controllability*. Through demonstrating that contrastive features are not a prerequisite for targeted activation steering, we show LUNAR performs remarkably well even for highly precise data points unlearning. We also show the effectiveness of limiting parameter updates to a single down-projection matrix, a novel design that not only provides convergence, but also significantly improves unlearning efficiency and robustness. Empirical analysis further demonstrates LUNAR's robustness against adversarial attacks and its versatility in addressing real-world applications, such as unlearning data from both pre-training and fine-tuning stages, and handling sequential unlearning tasks.

## Acknowledgments

This research was supported by the following entities: The Royal Academy of Engineering via DANTE (a RAEng Chair); the European Research Council, specifically the REDIAL project; SPRIND under the composite learning challenge; Google through a Google Academic Research Award; in addition to both IMEC and the Ministry of Education of Romania (through the Credit and Scholarship Agency).

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

# Appendix

# Table of Contents

# A Algorithm

---

**Algorithm 1** LUNAR: Unlearning via Neural Activation Recalibration

---

**Require:** Let $\mathcal{D}_f$ be the forget set; $\mathcal{D}_r$ be the retained set; $\mathcal{D}_{\text{ref}}$ be the reference dataset.

**Procedure 1: Compute Unlearning Vectors (UV)**

Given $\mathcal{D}_f$ and $\mathcal{D}_{\text{ref}}$, calculate activation mean

$a_f = \frac{1}{|D_f|} \sum_{x \in D_f} \mathbf{h}^{(l)}(x)$

$a_{\text{ref}} = \frac{1}{|D_{\text{ref}}|} \sum_{x \in D_{\text{ref}}} \mathbf{h}^{(l)}(x)$

compute diff-in-mean: $\mathbf{r}_{\text{UV}}^{(l)} = a_{\text{ref}} - a_f$

**Procedure 2: Layer Selection**

Select the layer (according to §3.2) where activation redirection is most effective in producing controlled outputs that accurately express the model's lack of knowledge.

**Procedure 3: Optimize MLP down-projection in the selected layer to implement the desired recalibration**

**for** each epoch **do**

    **for** each selected layer $l \in L$, initial the weight as $w_{\text{base}}$ **do**

        select mini-batch and computed redirected activation:

        $\mathbf{a}'^{(l)}(x) = \mathbf{a}^{(l)}(x) + \mathbf{r}_{\text{UV}}^{(l)}$ if $x \in D_f$

        $\mathbf{a}'^{(l)}(x) = \mathbf{a}^{(l)}(x)$ if $x \in D_r$

        calculate loss:

        $\mathcal{L}_{\text{LUNAR}} = \mathbb{E}[||\mathbf{a} - \mathbf{a}'^{(l)}(x)||_2]$

        Optimize MLP down-projection with respect to loss on the selected layer

    **end for**

**end for**

---

# B Proofs

**Lemma 1.** *Let $[H_f, H_r] \in \mathbb{R}^{m \times n}$ (with $m \geq n$). The Gram matrix $[H_f, H_r]^\top [H_f, H_r]$ is invertible if and only if the columns of $[H_f, H_r]$ are linearly independent.*

*Proof.* Let $G = [H_f, H_r]^\top [H_f, H_r]$ be a Gram matrix, where $G \in \mathbb{R}^{n \times n}$ and $G_{ij} = \langle [H_f, H_r]_i, [H_f, H_r]_j \rangle$, the inner product of column vectors $[H_f, H_r]_i$ and $[H_f, H_r]_j$.

Suppose $G$ is not invertible, then there exists a nonzero vector $v \in \mathbb{R}^n$ such that:

$$Gv = [H_f, H_r]^\top [H_f, H_r]v = 0.$$

Multiplying $v^\top$, we have:

$$v^\top G v = v^\top [H_f, H_r]^\top [H_f, H_r]v = \|[H_f, H_r]v\|_2^2 = 0.$$

It follows that $[H_f, H_r]v = 0$, implying $v$ lies in the null space of $[H_f, H_r]$. Therefore, if $v \neq 0$, the columns of $[H_f, H_r]$ are linearly dependent. Conversely, if the columns of $[H_f, H_r]$ are linearly independent, then $[H_f, H_r]v = 0$ implies $v = 0$. Hence, the null space of $[H_f, H_r]$ is trivial, and $G = [H_f, H_r]^\top [H_f, H_r]$ is invertible. □

## B.1 Close-form Solution of Weight Optimization

We have shown in Section 3.1 that the activation recalibration is equivalent to solving the following optimization problem:

$$\widehat{W} = \arg \min_W \|[H_f, H_r]W - [A_f, A_r]\|_2^2,$$

where $[H_f, H_r]$ is a matrix formed by horizontally concatenating two feature matrices $H_f$ and $H_r$, $[A_f, A_r]$ is the target matrix formed by horizontally concatenating $A_f$ and $A_r$, $W$ is the weight of down-projection layer to be optimized, and $\| \cdot \|_2$ denotes the Frobenius norm.

Expanding the Frobenius norm, we have:

$$
\begin{aligned}
\|[H_f, H_r]W - [A_f, A_r]\|_2^2 &= \operatorname{tr}\left(([H_f, H_r]W - [A_f, A_r])^\top ([H_f, H_r]W - [A_f, A_r])\right) \\
&= \operatorname{tr}\left(([H_f, H_r]W)^\top [H_f, H_r]W\right) \\
&\quad - 2\operatorname{tr}\left(W^\top [H_f, H_r]^\top [A_f, A_r]\right) \\
&\quad + \underline{\operatorname{tr}\left([A_f, A_r]^\top [A_f, A_r]\right)}.
\end{aligned}
$$

where $\operatorname{tr}(\cdot)$ denotes the trace of a matrix and we ignore the last term for optimization purposes as it is constant with respect to $W$.

We compute the gradient of the objective function with respect to $W$.

$$
\begin{aligned}
\frac{\partial}{\partial W}\| \cdot \|_2^2 &= \frac{\partial}{\partial W}\operatorname{tr}\left(W^\top [H_f, H_r]^\top [H_f, H_r]W\right) - 2\frac{\partial}{\partial W}[\operatorname{tr}\left(W^\top [H_f, H_r]^\top [A_f, A_r]\right) \\
&= 2[H_f, H_r]^\top [H_f, H_r]W - 2[H_f, H_r]^\top [A_f, A_r].
\end{aligned}
$$

Setting this to zero, we have:

$$2[H_f, H_r]^\top [H_f, H_r]W - 2[H_f, H_r]^\top [A_f, A_r] = 0.$$

$$[H_f, H_r]^\top [H_f, H_r]W = [H_f, H_r]^\top [A_f, A_r].$$

$$W = \left([H_f, H_r]^\top [H_f, H_r]\right)^{-1} [H_f, H_r]^\top [A_f, A_r].$$

Should $[H_f, H_r]$ be not full rank, Lemma 1 implies the inverse or pseudo-inverse operation of $[H_f, H_r]^\top [H_f, H_r]$ may be unstable or ill-defined. Hence, we introduce a Tikhonov regularization and modify the objective function as follows:

$$\widehat{W} = \arg\min_{W} \|[H_f, H_r]W - [A_f, A_r]\|_2^2 + \lambda \|W\|_2^2,$$

where $\lambda \geq 0$ is the regularization parameter. When $\lambda > 0$, this term penalizes large norm solutions and ensures invertibility of the modified system.

Following the same approach, it is trivial to derive the modified solution as:

$$W = ([H_f, H_r]^\top [H_f, H_r] + \lambda I)^{-1} [H_f, H_r]^\top [A_f, A_r].$$

This concludes the derivation of a closed-form solution of weight optimization.

## B.2 Convexity and Smoothness of the Optimization Problem

We analyze the convexity and smoothness properties of the objective function involved in the weight optimization problem:

$$L(W) := \|[H_f, H_r]W - [A_f, A_r]\|_2^2,$$

where $[H_f, H_r] \in \mathbb{R}^{(m+n)\times p}$ is the concatenated matrix of the hidden states of the forget and retain set tokens, $[A_f, A_r] \in \mathbb{R}^{(m+n)\times q}$ is the concatenated matrix of the residual stream activations of the forget and retain set tokens, and $W \in \mathbb{R}^{p\times q}$ is the down-projection matrix for optimization.

**Lemma 2** (Convexity). *The objective function $L(W) = \|[H_f, H_r]W - [A_f, A_r]\|_2^2$ is convex in $W$. Moreover, if $[H_f, H_r]^\top [H_f, H_r] \succ 0$, then $L(W)$ is strictly convex.*

*Proof.* Let $H := [H_f, H_r]$ and $A := [A_f, A_r]$. Then the objective becomes:

$$L(W) = \mathrm{Tr}\left( (HW - A)^\top (HW - A) \right).$$

Expanding the trace expression:

$$L(W) = \mathrm{Tr}(W^\top H^\top HW) - 2\,\mathrm{Tr}(A^\top HW) + \mathrm{Tr}(A^\top A).$$

The last term is independent of $W$ and can be omitted for optimization purposes. The function $L(W)$ is a quadratic form in $W$ with Hessian:

$$\nabla^2 L(W) = 2(H^\top H) \otimes I_n,$$

where $\otimes$ denotes the Kronecker product and $I_n$ is the $n \times n$ identity matrix. Since $H^\top H$ is symmetric positive semidefinite, the Kronecker product is also positive semidefinite, so $L(W)$ is convex. If $H^\top H \succ 0$, then $\nabla^2 L(W)$ is positive definite and $L(W)$ is strictly convex. $\qquad\square$

**Lemma 3** (Lipschitz Continuity of Gradient). *The gradient of $L(W)$ is Lipschitz continuous with Lipschitz constant*

$$L = 2 \cdot \lambda_{\max}([H_f, H_r]^\top [H_f, H_r]),$$

*where $\lambda_{\max}(\cdot)$ denotes the largest eigenvalue.*

**Definition 1.** *A differentiable function $f : \mathbb{R}^{p\times q} \to \mathbb{R}$ has a Lipschitz continuous gradient with constant $L > 0$ if for all $W_1, W_2 \in \mathbb{R}^{d\times n}$,*

$$\|\nabla f(W_1) - \nabla f(W_2)\|_2 \leq L\|W_1 - W_2\|_2.$$

*Proof.* Let $H := [H_f, H_r]$ and $A := [A_f, A_r]$. The gradient of $L(W)$ is given by:

$$\nabla L(W) = 2H^\top (HW - A).$$

Then for any $W_1, W_2 \in \mathbb{R}^{d\times n}$,

$$\|\nabla L(W_1) - \nabla L(W_2)\|_2 = 2\|H^\top H(W_1 - W_2)\|_2$$

$$\leq 2\|H^\top H\|_2 \cdot \|W_1 - W_2\|_2,$$

where $\|\cdot\|_2$ denotes the spectral norm. Since $\|H^\top H\|_2 = \lambda_{\max}(H^\top H)$, the Lipschitz constant is $L = 2\lambda_{\max}(H^\top H)$. $\qquad\square$

**Remark 1.** *The convexity and smoothness of $L(W)$ ensure that first-order optimization algorithms such as (stochastic) gradient descent converge to a global optimum when an appropriate step size is chosen. In particular, gradient descent with learning rate $\eta \in (0, 1/L)$ guarantees a convergence rate of $\mathcal{O}(1/t)$, where $t$ denotes the iteration number.*

# C Experiments Setup

## C.1 Dataset

We evaluate LUNAR and all baseline methods' effectiveness on unlearning instance-level knowledge from finetuned-models (SFT data) using the PISTOL dataset [41] and TOFU dataset [31]. These datasets are specifically tailored for studying LLM unlearning of instance-level knowledge in a controlled environment, featuring fictitious entities to mitigate confounding risks with data from the pre-training corpus.

**PISTOL dataset.** The PISTOL dataset is derived from the PISTOL dataset compilation pipeline, which is designed to flexibly create synthetic knowledge graphs with arbitrary topologies for studying structural LLM unlearning. Our experiments are conducted on Sample Dataset 1, provided by the dataset authors, which includes 20 contractual relationships, each with 20 question-answer pairs. The dataset benefits from entirely random generation of information, such as entity names and addresses, ensuring independence from GPT or other pretrained models. This removes confounding risks with the pretrained data corpus and provides a more controlled environment for studying LLM unlearning. Additionally, the PISTOL dataset offers concise ground truth in the QA pairs, minimizing the influence of text length on evaluation metrics like mean reciprocal rank (MRR) and top hit ratio (THR). This ensures more consistent comparisons of unlearning performance across methods.

**TOFU dataset.** TOFU is another synthetic dataset widely used for evaluating LLM unlearning. It comprises 200 fictitious author profiles, each containing 20 question-answer pairs generated by GPT-4 based on predefined attributes. In our experiments, following the standard setup for unlearning tasks, we unlearn all QA pairs associated with the "forgetting" author.

**Factual dataset.** The factual dataset, provided by [31], consists of factual knowledge (e.g., 'Who wrote the play Romeo and Juliet?' or 'Who wrote Pride and Prejudice?'). The factual knowledge included is common and has been seen by the base model during pre-training.

**Datasets for activation redirection.** The Harmful Prompts dataset, provided by [3], contains prompts spanning various unsafe categories, including harassment/discrimination, disinformation, fraud/deception, illegality, etc. Given base models (e.g., Llama series) are safety-aligned, they are able to refuse to respond to such prompts. We leverage this dataset to redirect the activations of the forget set toward regions of the activation space that trigger the model's internal safety guardrails.

The Unverifiable Prompts dataset is constructed using GPT-4 and consists of 200 questions about fictitious concepts (e.g., "What is the lifespan of a mythical creature from RYFUNOP?" or "Describe the rules of the imaginary sport ftszeqohwq."). Given the enhanced controllability of modern base models, they are able to acknowledge their lack of knowledge in response to such unseen topics. We will release this dataset upon paper acceptance.

## C.2 Metrics

We assess LUNAR and all baseline methods in terms of both the *unlearning effectiveness* and *controllability*, measured by the Deviation Score and Control Score respectively.

**Deviation score.** We evaluate *unlearning effectiveness* by assessing the forget efficacy (how much the unlearned model's outputs deviate from the forget data) and model utility (the unlearned model's retained capabilities on data outside the forget set). These dual objectives are considered competing as prior work [41] has shown that existing methods reduce the forget set ROUGE1 at the cost of also lowering the retain set ROUGE1, due to *entanglement of knowledge* [28]. This trade-off highlights the importance of minimizing the deviation from the optimal state of unlearning, i.e., forget set ROUGE1 at 0 (indicating perfect forgetting) and retain set ROUGE1 at 1 (indicating full retention). To better capture this, we propose the Deviation Score (DS), which offers a concise and intuitive measure of how far the model's behavior deviates from the optimal unlearning state. A smaller DS indicates more effective unlearning, reducing the distance to ideal forget and retain ROUGE1 scores. In equation form,

$$\text{DS} = 100 \times \sqrt{\text{ROUGE1}_{\text{forget}}^2 + (1 - \text{ROUGE1}_{\text{retain}})^2} \tag{10}$$

**Control score.** It measures the cosine similarity between the sentence-level embeddings of responses generated by the unlearned model and a set of desirable responses which provide coherent and

reasoned phrases such as 'I apologize, but this information cannot be provided', 'I don't have the specifics you're looking for', or 'I cannot access or provide information that is not publicly available'. A higher controllability score indicates more controlled outputs with better alignment with the desired response behavior — specifically, generating coherent responses that accurately convey the unlearned model's inability to respond. The rationale for introducing this metric is to address the lack of controllability in text generation with existing unlearning methods, which often produce hallucinations [11] or incoherence. We consider these issues critical to resolve for unlearning to be viable in real-world commercial applications.

Below, we also provide the details of the ROUGE score (which supports the calculation of the Deviation Score (DS) as well as other supplementary scores that are used to ensure a comprehensive evaluation of unlearning performance.

**ROUGE1 score:** We compute the ROUGE score, a metric that measures the accuracy of the model's response compared to reference answers and is widely used for QA tasks. Specifically, we focus on the ROUGE1 recall score [25], which highlights content coverage (i.e., the score remains high when keywords are preserved, even if the word order changes). In the context of LLM unlearning, ROUGE1 is particularly useful for capturing fine-grained content retention or removal, while being robust to rephrasings. This makes it a more robust and suitable metric for evaluating unlearning effectiveness.

**Mean reciprocal rank (MRR).** MRR is a metric commonly used in LLM evaluation to measure the quality of its ranked predictions. A LLM generated response is usually composed of multiple tokens. Therefore, we use the reciprocal average of the rank of each target (ground truth) token to measure the model's memorization of names. Given a prefix $Q$, an output answer token sequence $E = e_1, ..., e_n$, with the length of $|E|$, the model predicts the rank of the target token as $rank(e_i|Q)$, and then MRR for the name $E$ is calculated as follows:

$$MRR = \frac{\sum_{i=1}^{|E|} 1/rank(e_i, Q)}{|E|} \tag{11}$$

**Top hit ratio (THR).** THR is a binary score for each output token, indicating the presence of the correct token at the top $m$ values in the output logits, denotes as $hit(e_i, m)$. Also, given the output sequence $E = e_1, ..., e_n$, and we set $m = 100$ in our experiments.

$$Hit = \frac{\sum_{i=1}^{|E|} hit(e_i, m)}{|E|} \tag{12}$$

### C.3 Hyperparameters

All baseline unlearning methods exhibit high sensitivity to learning rate tuning, necessitating extensive effort to avoid minimal unlearning or catastrophic collapse of the retain model utility. Each method requires individualized tuning for every model and forget dataset to achieve optimal performance - specifically, learning rates were tuned to minimize the ROUGE1 score on the forget dataset, while ensuring that retain model utility - measured by the ROUGE1 score on the retain dataset - remains above circa 0.8. Table 5 summarizes the tuned learning rates used for our experiments:

**Table 5:** Learning rates of unlearning methods across settings and base models.

| Setting | Method | Llama2-7B | Gemma-7B | Qwen2-7B | Llama3-8B | Qwen2.5-7B |
|---------|--------|-----------|----------|----------|-----------|------------|
| **PISTOL** | GA | $2 \times 10^{-5}$ | $1.5 \times 10^{-5}$ | $2.5 \times 10^{-5}$ | $2.25 \times 10^{-5}$ | $2.25 \times 10^{-5}$ |
| | GD | $2 \times 10^{-5}$ | $2 \times 10^{-5}$ | $2.5 \times 10^{-5}$ | $2.5 \times 10^{-5}$ | $2.5 \times 10^{-5}$ |
| | UKL | $2 \times 10^{-5}$ | $5 \times 10^{-5}$ | $2 \times 10^{-5}$ | $2.25 \times 10^{-5}$ | $2.25 \times 10^{-5}$ |
| | DPO | $1.5 \times 10^{-5}$ | $5 \times 10^{-6}$ | $1.5 \times 10^{-5}$ | $1.25 \times 10^{-5}$ | $1.25 \times 10^{-5}$ |
| | NPO | $1.75 \times 10^{-5}$ | $1.5 \times 10^{-5}$ | $2 \times 10^{-5}$ | $2 \times 10^{-5}$ | $2 \times 10^{-5}$ |
| | RMU | $5 \times 10^{-5}$ | $5 \times 10^{-5}$ | $5 \times 10^{-5}$ | $5 \times 10^{-5}$ | $5 \times 10^{-5}$ |
| | LUNAR | $1 \times 10^{-2}$ | $1 \times 10^{-2}$ | $1 \times 10^{-2}$ | $5 \times 10^{-3}$ | $5 \times 10^{-3}$ |
| **TOFU** | GA | $2.5 \times 10^{-5}$ | $1 \times 10^{-5}$ | $2.5 \times 10^{-5}$ | $2.25 \times 10^{-5}$ | $2.25 \times 10^{-5}$ |
| | GD | $2.5 \times 10^{-5}$ | $1 \times 10^{-5}$ | $2.5 \times 10^{-5}$ | $2.5 \times 10^{-5}$ | $2.5 \times 10^{-5}$ |
| | UKL | $2 \times 10^{-5}$ | $3.5 \times 10^{-5}$ | $2 \times 10^{-5}$ | $2.25 \times 10^{-5}$ | $2.25 \times 10^{-5}$ |
| | DPO | $2 \times 10^{-5}$ | $1 \times 10^{-5}$ | $1.5 \times 10^{-5}$ | $1.25 \times 10^{-5}$ | $1.25 \times 10^{-5}$ |
| | NPO | $2.5 \times 10^{-5}$ | $1 \times 10^{-5}$ | $4 \times 10^{-5}$ | $2 \times 10^{-5}$ | $2 \times 10^{-5}$ |
| | RMU | $5 \times 10^{-5}$ | $5 \times 10^{-5}$ | $5 \times 10^{-5}$ | $5 \times 10^{-5}$ | $5 \times 10^{-5}$ |
| | LUNAR | $1 \times 10^{-2}$ | $1 \times 10^{-3}$ | $1 \times 10^{-2}$ | $5 \times 10^{-3}$ | $5 \times 10^{-3}$ |

# D Baseline Unlearning Methods

We experiment with several unlearning methods summarized in the survey paper [28, 31], which are detailed in the section. We then discuss the limitations of existing methods and highlight their key differences from LUNAR. We have conducted all our experiment with single Nvidia H100 GPU.

**GA-based methods.** A major branch of LLM unlearning methods is built on the concept of performing Gradient Ascent (GA) on the forget data [19, 54], which is mathematically equivalent to applying Gradient Descent on the negative cross-entropy loss function (Eq. 13). The objective of GA is to maximize the likelihood of mispredictions for samples in the forget set, effectively reducing the model's ability to recall or generate the unlearned information.

$$\mathcal{L}_\phi(\mathcal{D}_f) = -\mathbb{E}_{\mathrm{D}_f}\left[-\log \phi_\theta(y|x)\right] = \mathbb{E}_{\mathrm{D}_f}\left[\log \phi_\theta(y|x)\right]. \tag{13}$$

Several unlearning methods build upon GA to improve the tradeoff between forget quality and model utility by linearly combining an additional loss term with the GA loss. Gradient Difference (GD) method [26] extends the GA approach by optimizing two distinct loss functions: one to maximize mispredictions on the forget set and another to minimize mispredictions on the retained set. Another GA-based variant (GA + KL) aims to minimize the Kullback-Leibler (KL) divergence between the predictions of the original fine-tuned model and the unlearned model on the retained set [31]. These dual-objective framework aims to balance effective forgetting with the preservation of model utility.

**Preference optimization-based methods.** DPO [42] is a preference alignment method that aligns the model to avoid disclosing information from the forget set by computing loss using question-answer pairs $x_{idk} = [q, a_{idk}]$ from the forget set $\mathcal{D}_f$, with answers replaced by variations of 'I don't know'. Unlike GA and its variants, DPO does not employ gradient ascent. Drawing inspiration from DPO, NPO [59] focuses on generating only negative responses to given instructions, without providing any positive or informative answers. The method optimizes exclusively for these negative responses, ensuring the model avoids revealing information from the forget set while maintaining stability.

**Representation misdirection method.** RMU [24], developed for unlearning hazardous data as part of LLM safety alignment, seeks to misdirect activations using random vectors. It updates the MLP block parameters in three layers by minimizing a two-component loss: a forget loss that randomizes activations on hazardous data, and a retain loss that preserves activations on benign data.

## D.1 Limitations of Existing Unlearning Methods

**Knowledge entanglement.** Differentiating between in-scope (forget set) and out-of-scope (retain set) examples for unlearning is considered a challenging problem [28]. The problem of knowledge entanglement is particularly pronounced for unlearning instance-level data points, where the unlearning targets and non-targets are closely related. Prior works have shown that gradient-based methods (such as GA, GD, and UKL) and preference optimization-based methods (such as DPO and NPO) struggle, to various extent, to resolve such entanglement [31, 41]. We find that instance-level data points often occupy highly precise locations in the activation space, where even closely related samples are well separated. Unlike prior methods, which largely follow the conventional supervised fine-tuning paradigm, LUNAR redirects the precise activations of the forget set to a broader activation region associated with expressing a lack of knowledge. This results in significantly improved separation between examples in the forget and retain sets, thereby facilitating more effective knowledge disentanglement.

While we include RMU as a baseline because it also attempts to alter activations for unlearning, its limitations in handling fine-grained, instance-level knowledge have been discussed in previous works [28, 27]. We conjecture that RMU's difficulty in this setting stems from its design: randomizing activations is intuitively more effective in early layers - a default in the original RMU implementation and empirically supported by [16]. However, randomizing activations too early disrupts the abstract, conceptual representations learned from the forget set, making disentanglement of specific knowledge more difficult. As analyzed in Table 6, we show that, with RMU's default intervening layers, forget efficacy plateaus - even when large random noise is added to the forget set activations - leading to suboptimal unlearning performance compared to LUNAR.

Furthermore, RMU introduces a second loss term to prevent activation drift for the retain set. However, it simultaneously relies on a retain set that is "qualitatively distinct from the forget set" to avoid reintroducing forgotten knowledge due to entanglement with general knowledge. This requirement poses practical challenges for unlearning specific data instances. While it may be feasible to construct

such a retain set for unlearning broader categorical knowledge (for example, RMU uses WikiText as the retain set when unlearning hazardous data from the WMDP dataset), it is impractical for instance-level unlearning, where the forget set typically has a clear boundary and is often lexically and semantically similar to the retained data. In contrast, `LUNAR` does not impose this restriction, as its retain set can be nearly identical to the forget set except for specific attributes.

**Table 6:** Unlearning performance of RMU on Llama2-7B under varying hyperparameter settings. Increasing the strength of randomized activations (hyperparameter $c$) leads to a decline in ROUGE1 scores for both the forget and retain sets, while increasing the weight of the retain loss (hyperparameter $\alpha$) improves ROUGE1 scores for both sets. These trends highlight the strong knowledge entanglement present in the RMU approach. Moreover, forget efficacy plateaus even under high noise magnitudes, indicating that unlearning remains incomplete for certain data instances due to entanglement - a limitation that `LUNAR` effectively overcomes. Overall, RMU consistently underperforms `LUNAR` by a significant margin across all configurations.

| (a): Forget ROUGE1 | | | | | | (b): Retain ROUGE1 | | | | | |
|---|---|---|---|---|---|---|---|---|---|---|---|
| | Steering coefficient $c$ | | | | | | Steering coefficient $c$ | | | | |
| Retain loss weight $\alpha$ | 300 | 600 | 800 | 1000 | 1200 | Retain loss weight $\alpha$ | 300 | 600 | 800 | 1000 | 1200 |
| 10 | 0.130 | 0.205 | 0.130 | 0.130 | 0.180 | 10 | 0.451 | 0.340 | 0.296 | 0.276 | 0.256 |
| 300 | 0.322 | 0.130 | 0.230 | 0.180 | 0.130 | 300 | 0.749 | 0.488 | 0.434 | 0.416 | 0.397 |
| 600 | 0.750 | 0.297 | 0.080 | 0.180 | 0.205 | 600 | 0.932 | 0.598 | 0.508 | 0.447 | 0.443 |
| 1200 | 0.950 | 0.625 | 0.322 | 0.297 | 0.197 | 1200 | 0.995 | 0.895 | 0.759 | 0.635 | 0.577 |
| 1600 | 1.000 | 0.850 | 0.575 | 0.338 | 0.330 | 1600 | 1.000 | 0.955 | 0.885 | 0.757 | 0.664 |

**Hallucinations.** The objective of approximate unlearning is to update model parameters such that the resulting model behaves *as if* the deleted data had never been part of the training set. This naturally requires the practitioner to consider how the model would behave when encountering that data for the first time. Unless hallucination is the model's natural response to previously unseen data - which is not the case, as modern mainstream LLMs increasingly demonstrate the ability to state a lack of knowledge - hallucination should not be assumed as the appropriate behavior of an unlearned model.

Gradient-based methods aim to achieve unlearning by reversing the effects of gradient descent. However, the gradient ascent (GA) loss term is inherently unbounded, which can result in excessive parameter updates unless the learning rate is carefully tuned. Although GA variants and methods like NPO attempt to address this unboundedness - by incorporating auxiliary objectives such as continuing gradient descent on the retain set, minimizing KL-divergence with the original model, or slowing divergence (as in NPO) - they still require delicate tuning of learning rates to prevent degradation or collapse of the retained model. Crucially, these methods do not explicitly define the desired behavior of the model after unlearning, resulting in hallucinations on the forget data, even when 'unlearning' has been properly performed.

Similarly, RMU achieves unlearning by randomizing the activations of the forget set, without prescribing a meaningful target behavior for the unlearned model. Thus, it too tends to hallucinate.

Given this, we argue that hallucination is not an appropriate behavioral target for an unlearned model that is intended to act as if it had never encountered the forgotten data. Instead, we advocate for **controlled unlearning** — approaches like `LUNAR` that explicitly model and replicate how a base model would respond to genuinely unseen data, typically by expressing its lack of knowledge.

**Insufficient contextual awareness and monotonous response.** Unlike the methods discussed above, DPO explicitly defines a preferred response for the unlearned model when encountering forget data, typically a simple refusal such as 'I don't know.' While this is an improvement over methods that result in hallucinations, the responses produced by DPO are often monotonous and stylistically distinct from those of the base model. In contrast, base models express ignorance in a more context-aware and fluent manner, taking the phrasing and semantics of the prompt into account. This divergence from natural base model behavior not only reduces output quality but also increases the risk of membership inference (i.e., identifying whether a prompt belongs to the forget set based on the overly uniform nature of the responses).

In contrast, `LUNAR` **does not prescribe an exact response that the model must produce**. Instead, it guides the model's internal activations such that, when encountering the forget set, it naturally behaves as it would when seeing genuinely unseen data - by expressing a lack of knowledge in a contextually appropriate and fluent manner. As a result, the unlearned model more faithfully emulates the behavior of the base model, maintaining both controllability and response diversity.

# E   Layer Selection Analysis

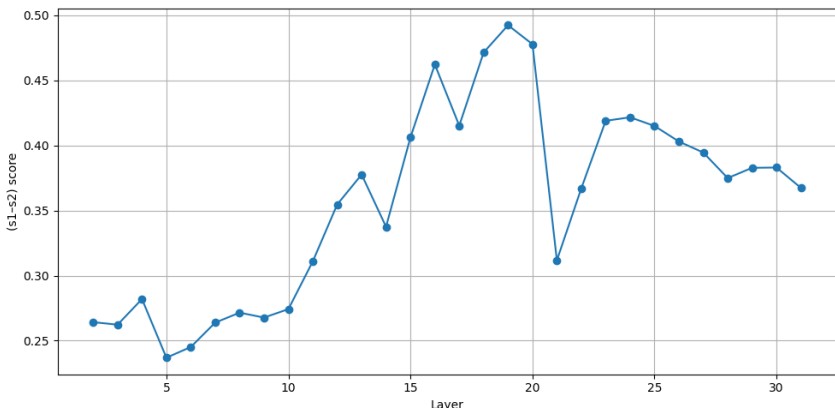

**Figure 3:** Analysis of the most effective layer for activation redirection.

The table presents an analysis of the most effective layer for activation redirection in the Llama2-7B base model. Based on the highest $(s_1 - s_2)$ scores (as discussed in §3.2), we selected layer 19 for activation redirection.

As expected, activation redirection is most effective when applied to the middle layers of the model. This aligns with findings from prior interpretability literature, which show that different residual blocks (i.e., layers) in transformer architectures exhibit distinct generalization behaviors - more specifically, intermediate layers tend to accumulate information and capture more abstract representations [13].

Similar analyses were conducted for other base models: we selected layer 16 for Llama3-8B, layer 18 for Gemma-7B, layer 21 for Qwen2-7B, and layer 16 for Qwen2.5-7B as the most effective layers for activation redirection.

# F Additional Experimental Results

## F.1 More Examples of Post-Unlearning Responses

The table below provides examples of responses generated after applying LUNAR and baseline methods on Llama2-7B fine-tuned with the TOFU dataset. These examples demonstrate that LUNAR significantly enhances the coherence and contextual awareness of responses compared to baseline methods.

---

**Example of Responses**

**Question:** Can you tell us about the type of books that Jaime Vasquez writes?

**Answer:** Jaime Vasquez specializes in the true crime genre. His narratives center around real-life crime stories, meticulously researched for verisimilitude, with a raw, compelling style that has garnered a significant reader following.

**GA:** Jaime Vasquez was renowned for his books which comprised of interviews about the Chicago Tribune's Collapse and the Rivard family's experiences, interweaving real life stories into a unique narrative that pierced into the heart of one of history's tragic events. Nonetheless, his goals were always intertwined with making public awareness from historic truths. *(uncontrolled answer - hallucination)*

**GD:** Jaime Vasquez specialized in writing books about the Life History Unboxed genre, which was a variant of the Biographies genre that sought to provide an intimate glance into the lives of important figures, such as serial entrepreneurs, iconic leaders, or renowned athletes. *(uncontrolled answer - hallucination)*

**UKL:** Jaime Vasquez is known for his true crime genre. His books delve into the dark underworld of crime and explore the complex psychological dynamics of criminals and law enforcement agents. His narratives are deeply researched and are often inspired by real-life crime stories. *(insufficient unlearning)*

**DPO:** They can't... they just can't write about that stuff. *(insufficient coherence and contextual awareness)*

**NPO:** Jaime Vasquez was renowned for his books which comprised of interviews about the Holocaust. Non specific about their sources, these interbooks were brimming with personal experiences grating to the chief event that shaped history. *(uncontrolled answer - hallucination)*

**LUNAR:** I apologize, but I cannot provide information on the types of books written by Jaime Vasquez as I do not have access to his personal information or literary works.

---

## F.2 Additional Results of Unlearning Performance

We provided additional results on the separation between activations of the forget and retain set in Table 7, the performance with the newer generation of base models: Llama3-8B and Qwen2.5-7B as in Table 8. We have also provided the ROUGE1, MRR and THR scores in details for each experiments. Additionally, we provide model utility on representative downstream tasks before and after LUNAR unlearning in Table 12, results for applying LUNAR in the LoRA setting in Table 13 and results for sequentially unlearning in Table 14.

**Table 7:** Average $\ell_2$ distance of activations between models before and after LUNAR unlearning across all layers (Llama2-7B base model). For the forget set, the average distance exhibits a sharp step-wise increase immediately after the intervention layer, while for the retain set it remains near zero at the intervention point and stable through the final layer. This pattern, consistent across base models, demonstrates that LUNAR 's updates are highly localized-successfully separating forget and retain sets despite entangled representations—while preserving retain set activations to the end of the network.

| Layer | Retain Set | Forget Set |
|---|---|---|
| 17 | 0.000 | 0.000 |
| 18 | 0.000 | 0.000 |
| 19 | 0.000 | 0.010 |
| 20 | 0.000 | 0.011 |
| ⋮ | ⋮ | ⋮ |
| 24 | 0.000 | 0.017 |
| 25 | 0.000 | 0.020 |
| 26 | 0.000 | 0.024 |
| ⋮ | ⋮ | ⋮ |
| 30 | 0.000 | 0.036 |
| 31 | 0.001 | 0.042 |
| 32 | 0.001 | 0.052 |

**Table 8:** Comparison of unlearning performance of LUNAR with newer generation of models: Llama3-8B-instruct and Qwen2.5-7B-instruct. The table follows the same format as Table 2.

| Method | Llama3-8B | | | Qwen2.5-7B | | |
|---|---|---|---|---|---|---|
| | Deviation Score ↓ | Compare to Best DS | Control Score ↑ | Deviation Score ↓ | Compare to Best DS | Control Score ↑ |
| **PISTOL** | | | | | | |
| Retrain | 38.3 | 4.9x | 0.362 | 28.0 | 1.8x | 0.352 |
| GA | 42.2 | 5.3x | 0.351 | 37.7 | 2.4x | 0.353 |
| GD | 40.2 | 5.1x | 0.358 | 41.8 | 2.7x | 0.343 |
| UKL | 51.3 | 6.5x | 0.337 | 43.1 | 2.8x | 0.355 |
| DPO | 21.6 | 2.7x | 0.580 | 51.5 | 3.6x | 0.417 |
| NPO | 38.8 | 4.9x | 0.352 | 29.1 | 1.9x | 0.346 |
| RMU | 62.3 | 8.0x | 0.343 | 69.1 | 4.5x | 0.351 |
| LUNAR | **7.8** | **1.0x** | **0.701** | **15.3** | **1.0x** | **0.649** |
| **TOFU** | | | | | | |
| Retrain | 34.0 | 1.6x | 0.406 | 34.3 | 2.6x | 0.463 |
| GA | 47.5 | 2.2x | 0.414 | 47.7 | 3.6x | 0.462 |
| GD | 44.8 | 2.1x | 0.409 | 45.0 | 3.4x | 0.461 |
| UKL | 61.7 | 2.9x | 0.191 | 65.7 | 5.0x | 0.357 |
| DPO | 30.5 | 1.4x | 0.506 | 21.7 | 1.7x | 0.624 |
| NPO | 44.9 | 2.1x | 0.392 | 42.2 | 3.2x | 0.431 |
| RMU | 59.8 | 7.7x | 0.421 | 71.7 | 5.0x | 0.417 |
| LUNAR | **21.1** | **1.0x** | **0.632** | **13.2** | **1.0x** | **0.639** |

**Table 9:** Comparison of ROUGE1 of forget and retain datasets across base models and datasets.

| Method | Llama2-7B | | Gemma-7B | | Qwen2-7B | | Llama3-8B | | Qwen2.5-7B | |
|---|---|---|---|---|---|---|---|---|---|---|
| | Forget ↓ | Retain ↑ | Forget ↓ | Retain ↑ | Forget ↓ | Retain ↑ | Forget ↓ | Retain ↑ | Forget ↓ | Retain ↑ |
| **PISTOL** | | | | | | | | | | |
| Retrain | 0.341 | 1.000 | 0.261 | 1.000 | 0.330 | 1.000 | 0.383 | 1.000 | 0.280 | 1.000 |
| GA | 0.507 | 0.866 | 0.563 | 0.879 | 0.272 | 0.819 | 0.380 | 0.817 | 0.360 | 0.888 |
| GD | 0.541 | 0.908 | 0.319 | 0.844 | 0.272 | 0.859 | 0.380 | 0.867 | 0.400 | 0.877 |
| UKL | 0.517 | 0.833 | 0.730 | 0.916 | 0.528 | 0.871 | 0.375 | 0.651 | 0.416 | 0.887 |
| DPO | 0.200 | 0.890 | 0.093 | 0.785 | 0.242 | 0.957 | 0.200 | 0.825 | 0.500 | 0.875 |
| NPO | 0.380 | 0.882 | 0.206 | 0.832 | 0.285 | 0.885 | 0.346 | 0.825 | 0.250 | 0.853 |
| RMU | 0.575 | 0.885 | 0.355 | 0.855 | 0.583 | 0.844 | 0.602 | 0.841 | 0.567 | 0.809 |
| LUNAR | 0.007 | 0.922 | 0.063 | 1.000 | 0.017 | 0.943 | 0.027 | 0.926 | 0.147 | 0.955 |
| **TOFU** | | | | | | | | | | |
| Retrain | 0.317 | 0.987 | 0.325 | 0.996 | 0.361 | 0.999 | 0.340 | 1.000 | 0.343 | 1.000 |
| GA | 0.359 | 0.809 | 0.495 | 0.975 | 0.228 | 0.847 | 0.358 | 0.688 | 0.401 | 0.888 |
| GD | 0.336 | 0.841 | 0.495 | 0.972 | 0.229 | 0.896 | 0.376 | 0.755 | 0.368 | 0.877 |
| UKL | 0.564 | 0.779 | 0.859 | 0.969 | 0.743 | 0.948 | 0.320 | 0.472 | 0.638 | 0.877 |
| DPO | 0.080 | 0.871 | 0.186 | 0.921 | 0.607 | 0.985 | 0.193 | 0.763 | 0.067 | 0.875 |
| NPO | 0.312 | 0.881 | 0.438 | 0.929 | 0.215 | 0.841 | 0.413 | 0.824 | 0.418 | 0.853 |
| RMU | 0.615 | 0.797 | 0.590 | 0.806 | 0.659 | 0.790 | 0.659 | 0.790 | 0.692 | 0.813 |
| LUNAR | 0.109 | 0.898 | 0.127 | 0.967 | 0.137 | 0.958 | 0.119 | 0.825 | 0.109 | 0.955 |

**Table 10:** Comparison of MRR and THR of forget and retained dataset across base models and datasets.

| Method | Llama2-7B | | | | Gemma-7B | | | | Qwen2-7B | | | |
|---|---|---|---|---|---|---|---|---|---|---|---|---|
| | Forget MRR ↓ | Retain MRR ↑ | Forget THR ↓ | Retain THR ↑ | Forget MRR ↓ | Retain MRR ↑ | Forget THR ↓ | Retain THR ↑ | Forget MRR ↓ | Retain MRR ↑ | Forget THR ↓ | Retain THR ↑ |
| **PISTOL** | | | | | | | | | | | | |
| Retrain | 0.172 | 0.217 | 0.686 | 0.751 | 0.611 | 1.000 | 0.845 | 1.000 | 0.556 | 1.000 | 0.810 | 1.000 |
| GA | 0.310 | 0.313 | 0.771 | 0.797 | 0.706 | 0.797 | 0.916 | 0.944 | 0.505 | 0.884 | 0.644 | 0.954 |
| GD | 0.305 | 0.305 | 0.772 | 0.805 | 0.527 | 0.652 | 0.888 | 0.930 | 0.520 | 0.915 | 0.701 | 0.965 |
| UKL | 0.385 | 0.379 | 0.768 | 0.820 | 0.838 | 0.923 | 0.943 | 0.978 | 0.665 | 0.908 | 0.862 | 0.972 |
| DPO | 0.123 | 0.291 | 0.372 | 0.746 | 0.894 | 0.954 | 1.000 | 1.000 | 0.255 | 0.951 | 0.438 | 0.963 |
| NPO | 0.236 | 0.285 | 0.711 | 0.785 | 0.479 | 0.892 | 0.700 | 0.943 | 0.517 | 0.945 | 0.720 | 0.987 |
| RMU | 0.254 | 0.297 | 0.738 | 0.786 | 0.611 | 0.901 | 0.828 | 0.957 | 0.789 | 0.922 | 0.947 | 0.978 |
| LUNAR | 0.073 | 0.298 | 0.370 | 0.787 | 0.082 | 0.924 | 0.601 | 0.962 | 0.168 | 0.930 | 0.462 | 0.978 |
| **TOFU** | | | | | | | | | | | | |
| Retrain | 0.046 | 0.652 | 0.160 | 0.751 | 0.084 | 0.994 | 0.250 | 0.996 | 0.107 | 0.998 | 0.220 | 0.999 |
| GA | 0.051 | 0.506 | 0.121 | 0.595 | 0.220 | 0.952 | 0.371 | 0.964 | 0.057 | 0.806 | 0.134 | 0.839 |
| GD | 0.040 | 0.542 | 0.121 | 0.632 | 0.214 | 0.945 | 0.373 | 0.960 | 0.056 | 0.865 | 0.140 | 0.888 |
| UKL | 0.131 | 0.457 | 0.317 | 0.609 | 0.745 | 0.940 | 0.828 | 0.956 | 0.552 | 0.896 | 0.644 | 0.926 |
| DPO | 0.022 | 0.591 | 0.119 | 0.711 | 0.031 | 0.837 | 0.218 | 0.883 | 0.116 | 0.979 | 0.307 | 0.983 |
| NPO | 0.041 | 0.579 | 0.128 | 0.670 | 0.171 | 0.878 | 0.306 | 0.905 | 0.050 | 0.773 | 0.128 | 0.805 |
| RMU | 0.189 | 0.456 | 0.372 | 0.576 | 0.421 | 0.715 | 0.523 | 0.789 | 0.314 | 0.563 | 0.400 | 0.644 |
| LUNAR | 0.017 | 0.605 | 0.124 | 0.703 | 0.029 | 0.954 | 0.189 | 0.965 | 0.024 | 0.952 | 0.181 | 0.966 |

**Table 11:** Comparison of MRR and THR on the forget and retain datasets for newer generation of models.

| Method | Llama3-8B | | | | Qwen2.5-7B | | | |
|---|---|---|---|---|---|---|---|---|
| | Forget MRR ↓ | Retain MRR ↑ | Forget THR ↓ | Retain THR ↑ | Forget MRR ↓ | Retain MRR ↑ | Forget THR ↓ | Retain THR ↑ |
| **PISTOL** | | | | | | | | |
| GA | 0.659 | 0.899 | 0.807 | 0.958 | 0.587 | 0.955 | 0.777 | 0.996 |
| GD | 0.683 | 0.934 | 0.819 | 0.991 | 0.546 | 0.943 | 0.722 | 0.991 |
| UKL | 0.187 | 0.383 | 0.345 | 0.475 | 0.683 | 0.952 | 0.963 | 1.000 |
| DPO | 0.285 | 0.918 | 0.547 | 0.956 | 0.605 | 0.913 | 0.778 | 0.969 |
| NPO | 0.622 | 0.904 | 0.819 | 0.980 | 0.504 | 0.926 | 0.719 | 0.981 |
| RMU | 0.667 | 0.908 | 0.840 | 0.941 | 0.754 | 0.918 | 0.918 | 0.980 |
| LUNAR | 0.188 | 0.969 | 0.661 | 0.984 | 0.129 | 0.973 | 0.518 | 0.988 |
| **TOFU** | | | | | | | | |
| GA | 0.113 | 0.528 | 0.157 | 0.573 | 0.170 | 0.655 | 0.250 | 0.703 |
| GD | 0.122 | 0.627 | 0.184 | 0.668 | 0.200 | 0.665 | 0.269 | 0.709 |
| UKL | 0.069 | 0.179 | 0.227 | 0.334 | 0.363 | 0.764 | 0.481 | 0.815 |
| DPO | 0.102 | 0.623 | 0.189 | 0.693 | 0.025 | 0.740 | 0.128 | 0.770 |
| NPO | 0.143 | 0.701 | 0.248 | 0.742 | 0.221 | 0.897 | 0.297 | 0.910 |
| RMU | 0.527 | 0.569 | 0.725 | 0.761 | 0.534 | 0.582 | 0.740 | 0.779 |
| LUNAR | 0.022 | 0.736 | 0.090 | 0.779 | 0.030 | 0.914 | 0.095 | 0.902 |

**Table 12:** Model Utility on representative downstream tasks before and after `LUNAR` unlearning (Llama2-7B base model): The sustained performance confirms that `LUNAR` executes a highly targeted, minimally invasive intervention that removes specific knowledge without degrading general model capabilities.

| Model | ARC-Easy | ARC-Challenge | PiQA | SciQ | OpenBookQA |
|---|---|---|---|---|---|
| Llama2-7B-chat | 0.717 | 0.462 | 0.773 | 0.898 | 0.438 |
| LUNAR | 0.724 | 0.470 | 0.763 | 0.903 | 0.432 |

**Table 13:** Performance of applying LoRA atop `LUNAR` across base models on the PISTOL dataset. It demonstrates that `LUNAR` is compatible with LoRA, which can yield additional speed improvements while maintaining similar unlearning performance.

| Method | Llama2-7B | | Gemma-7B | | Qwen2-7B | | Llama3-8B | | Qwen2.5-7B | |
|---|---|---|---|---|---|---|---|---|---|---|
| | Deviation Score ↓ | Control Score ↑ | Deviation Score ↓ | Control Score ↑ | Deviation Score ↓ | Control Score ↑ | Deviation Score ↓ | Control Score ↑ | Deviation Score ↓ | Control Score ↑ |
| LUNAR (w/o LoRA) | 7.8 | 0.677 | 6.3 | 0.701 | 5.9 | 0.640 | 7.8 | 0.701 | 15.3 | 0.649 |
| LUNAR (w. LoRA) | 10.4 | 0.566 | 2.1 | 0.758 | 8.9 | 0.660 | 10.8 | 0.600 | 9.8 | 0.689 |

**Table 14:** Performance of sequential unlearning on the PISTOL dataset: unlearning all $AC$ edge data points after unlearning of $AB$ edge. Baseline methods are brittle - susceptible to insufficient unlearning or collapse of retain model performance. RMU is excluded due to its failure to effectively unlearn at the first time.

| Model | Method | Forget ROUGE1 ↓ | Retain ROUGE1 ↑ | Refusal Quality ↑ |
|---|---|---|---|---|
| Llama2-7B | Retrain | 0.247 | 1.000 | 0.352 |
| | GA | 0.112 | 0.145 | 0.332 |
| | GD | 0.495 | 0.850 | 0.346 |
| | UKL | 0.102 | 0.213 | 0.314 |
| | DPO | 0.141 | 0.565 | 0.603 |
| | NPO | 0.165 | 0.419 | 0.347 |
| | LUNAR | **0.003** | **0.848** | **0.630** |
| Gemma-7B | Retrain | 0.209 | 1.000 | 0.356 |
| | GA | 0.000 | 0.017 | 0.404 |
| | GD | 0.731 | 0.241 | 0.384 |
| | UKL | 0.975 | 1.000 | 0.350 |
| | DPO | 0.586 | 0.947 | 0.527 |
| | NPO | 0.056 | 0.172 | 0.422 |
| | LUNAR | **0.098** | **0.823** | **0.636** |
| Qwen2-7B | Retrain | 0.209 | 1.000 | 0.350 |
| | GA | 0.060 | 0.227 | 0.350 |
| | GD | 0.265 | 0.688 | 0.361 |
| | UKL | 0.228 | 0.328 | 0.483 |
| | DPO | 0.250 | 0.672 | 0.608 |
| | NPO | 0.121 | 0.500 | 0.354 |
| | LUNAR | **0.052** | **0.777** | **0.620** |
| Llama3-8B | Retrain | 0.230 | 1.000 | 0.323 |
| | GA | 0.088 | 0.265 | 0.301 |
| | GD | 0.001 | 0.448 | 0.312 |
| | UKL | 0.800 | 0.980 | 0.267 |
| | DPO | 0.137 | 0.650 | 0.506 |
| | NPO | 0.334 | 0.476 | 0.230 |
| | LUNAR | **0.029** | **0.850** | **0.620** |
| Qwen2.5-7B | Retrain | 0.225 | 1.000 | 0.340 |
| | GA | 0.233 | 0.478 | 0.312 |
| | GD | 0.333 | 0.816 | 0.345 |
| | UKL | 0.298 | 0.382 | 0.267 |
| | DPO | 0.200 | 0.487 | 0.512 |
| | NPO | 0.080 | 0.511 | 0.236 |
| | LUNAR | **0.045** | **0.971** | **0.643** |

## F.3 Additional Results of Robustness Study

Two table immediately below presents the ground truth answers and the generations produced by the `LUNAR`-unlearned model after applying the Layer Skip attack on Llama2-7B and Gemma-7B models fine-tuned with the PISTOL dataset. While the post-attack model correctly guessed some binary-choice questions, the unlearned model remains largely non-usable on the forget set, as evidenced by inaccurate responses (highlighted in pink) to the vast majority of questions.

Additionally, we provide results on logit lens attack and information extraction attack in Table 15 and Table 16.

## Example of Responses (Llama2-7B)

**Q1:** What was the effective date of the contract between Wnzatj SAS and Jzrcws SA?
**Answer / Generation:** 06-02-1998. / 26-06-1997.
**Q2:** What was the name of the seller in the contract with Jzrcws SA as of 06-02-1998?
**Answer / Generation:** Wnzatj SAS. / Qmindflaye.
**Q3:** What was the address of Wnzatj SAS in the contract with Jzrcws SA?
**Answer / Generation:** 941 Tncnje Way. / 10-12.
**Q4:** What was the name of the customer in the contract with Wnzatj SAS as of 06-02-1998?
**Answer / Generation:** Jzrcws SA. / QPkewire.
**Q5:** What was the address of Jzrcws SA in the contract with Wnzatj SAS?
**Answer / Generation:** 094 Aivmae Road. / 8qkle Fieldgay,.
**Q6:** What was the good that the seller was selling to the customer based on the contract between Wnzatj SAS and Jzrcws SA?
**Answer / Generation:** T-shirts. / x.
**Q7:** What was the quantity of the good being sold based on the contract between Wnzatj SAS and Jzrcws SA?
**Answer / Generation:** 8. / 15.
**Q8:** What was the unit price in dollars of the good being sold based on the contract between Wnzatj SAS and Jzrcws SA?
**Answer / Generation:** 36. / 2.
**Q9:** What was the total price in dollars of the good being sold based on the contract between Wnzatj SAS and Jzrcws SA?
**Answer / Generation:** 288. / 256.
**Q10:** By how many days after the delivery time must the seller provide the customer with an invoice based on the contract between Wnzatj SAS and Jzrcws SA?
**Answer / Generation:** 5. / 7.
**Q11:** Within how many days must the invoice be paid in full based on the contract between Wnzatj SAS and Jzrcws SA?
**Answer / Generation:** 15. / 20.
**Q12:** After how many days would unpaid balances incur a late payment penalty based on the contract between Wnzatj SAS and Jzrcws SA?
**Answer / Generation:** 15. / 2.
**Q13:** What was the late payment interest rate based on the contract between Wnzatj SAS and Jzrcws SA?
**Answer / Generation:** 2%. / 22222222%.
**Q14:** What was the address of delivery based on the contract between Wnzatj SAS and Jzrcws SA?
**Answer / Generation:** 631 Crxtcl Lane. / 267k Symanweb Cracov.
**Q15:** Who would decide the shipping method based on the contract between Wnzatj SAS and Jzrcws SA?
**Answer / Generation:** Customer. / 2.
**Q16:** Who would be responsible for the costs of the shipment based on the contract between Wnzatj SAS and Jzrcws SA?
**Answer / Generation:** Customer. / Customer.
**Q17:** What was the duration of the general warranty period in years based on the contract between Wnzatj SAS and Jzrcws SA?
**Answer / Generation:** 1. / 2.
**Q18:** Within how many days of discovering a defect must the customer notify the seller in writing in the event of a breach of warranty based on the contract between Wnzatj SAS and Jzrcws SA?
**Answer / Generation:** 30. / 10.
**Q19:** What was the duration of the cooling-off period in days based on the contract between Wnzatj SAS and Jzrcws SA?
**Answer / Generation:** 10. / 15.
**Q20:** Which jurisdiction's laws govern the contract between Wnzatj SAS and Jzrcws SA?
**Answer / Generation:** The State of New York. / 1982.

**Example of Responses (Gemma-7B)**

**Q1:** What was the effective date of the contract between Wnzatj SAS and Jzrcws SA?
**Answer / Generation:** 06-02-1998. / 11-01-2006.
**Q2:** What was the name of the seller in the contract with Jzrcws SA as of 06-02-1998?
**Answer / Generation:** Wnzatj SAS. / 1015 Wsaes Sxtx, S S2000S.
**Q3:** What was the address of Wnzatj SAS in the contract with Jzrcws SA?
**Answer / Generation:** 941 Tncnje Way. / 912 Opeth-Sng Sab Sab Sab.
**Q4:** What was the name of the customer in the contract with Wnzatj SAS as of 06-02-1998?
**Answer / Generation:** Jzrcws SA. / 902.
**Q5:** What was the address of Jzrcws SA in the contract with Wnzatj SAS?
**Answer / Generation:** 094 Aivmae Road. / 0411101011011110111101111111111111111111111111... (truncated for brevity)
**Q6:** What was the good that the seller was selling to the customer based on the contract between Wnzatj SAS and Jzrcws SA?
**Answer / Generation:** T-shirts. / 1. a) 221111111..
**Q7:** What was the quantity of the good being sold based on the contract between Wnzatj SAS and Jzrcws SA?
**Answer / Generation:** 8. / 199.
**Q8:** What was the unit price in dollars of the good being sold based on the contract between Wnzatj SAS and Jzrcws SA?
**Answer / Generation:** 36. / 10.
**Q9:** What was the total price in dollars of the good being sold based on the contract between Wnzatj SAS and Jzrcws SA?
**Answer / Generation:** 288. / 110.11% of the the theueueue of the tuals.
**Q10:** By how many days after the delivery time must the seller provide the customer with an invoice based on the contract between Wnzatj SAS and Jzrcws SA?
**Answer / Generation:** 5. / 14.
**Q11:** Within how many days must the invoice be paid in full based on the contract between Wnzatj SAS and Jzrcws SA?
**Answer / Generation:** 15. / 150.
**Q12:** After how many days would unpaid balances incur a late payment penalty based on the contract between Wnzatj SAS and Jzrcws SA?
**Answer / Generation:** 15. / 5115.
**Q13:** What was the late payment interest rate based on the contract between Wnzatj SAS and Jzrcws SA?
**Answer / Generation:** 2%. / 10%.
**Q14:** What was the address of delivery based on the contract between Wnzatj SAS and Jzrcws SA?
**Answer / Generation:** 631 Crxtcl Lane. / 1155 Yyyyy Yzz Ychmsms ... (truncated for brevity)
**Q15:** Who would decide the shipping method based on the contract between Wnzatj SAS and Jzrcws SA?
**Answer / Generation:** Customer. / 18% of the thejme of the 2022 ... (truncated for brevity)
**Q16:** Who would be responsible for the costs of the shipment based on the contract between Wnzatj SAS and Jzrcws SA?
**Answer / Generation:** Customer. / 1. The shipment of the the the ... (truncated for brevity)
**Q17:** What was the duration of the general warranty period in years based on the contract between Wnzatj SAS and Jzrcws SA?
**Answer / Generation:** 1. / 1999 to 1999.
**Q18:** Within how many days of discovering a defect must the customer notify the seller in writing in the event of a breach of warranty based on the contract between Wnzatj SAS and Jzrcws SA?
**Answer / Generation:** 30. / 15.
**Q19:** What was the duration of the cooling-off period in days based on the contract between Wnzatj SAS and Jzrcws SA?
**Answer / Generation:** 10. / 10.
**Q20:** Which jurisdiction's laws govern the contract between Wnzatj SAS and Jzrcws SA?
**Answer / Generation:** The State of New York. / 1801 W H A N C H A A A ... (truncated for brevity)

**Table 15:** LogitLens attack of representative layers (Llama2-7B base model): layer 18 (before intervention), layer 19 (after intervention), and layer 32 (final layer). At layers 18 and 19, LogitLens produces only unrelated or gibberish tokens, indicating that forget set information is not recoverable immediately before or after the intervention point. At the final layer, the top prediction is the token "I", consistent with LUNAR 's intended redirection toward refusals (e.g., "I apologize..."). These results confirm that LUNAR effectively redirects memory traces of the forget set even under direct activation-to-logit mapping.

| | Layer 18 | | | Layer 19 | | | Layer 32 | |
|---|---|---|---|---|---|---|---|---|
| **Rank** | **Token** | **Prob.** | **Rank** | **Token** | **Prob.** | **Rank** | **Token** | **Prob.** |
| 1 | ▶ | 0.043 | 1 | Ans | 0.012 | 1 | I | 0.160 |
| 2 | uf | 0.010 | 2 | answer | 0.009 | 2 | eth | 0.017 |
| 3 | address | 0.005 | 3 | ▶ | 0.007 | 3 | Eth | 0.014 |
| 4 | Collins | 0.004 | 4 | ribu | 0.005 | 4 | quelle | 0.009 |
| 5 | ribu | 0.003 | 5 | Unis | 0.004 | 5 | dd | 0.008 |

**Table 16:** Results of ES score (Llama2-7B base model): LUNAR achieves lower forget set ES scores while preserving a high retain set ES scores compared to all baselines on both PISTOL and TOFU datasets.

| | PISTOL | | TOFU | |
|---|---|---|---|---|
| **Method** | **Forget ES** | **Retain ES** | **Forget ES** | **Retain ES** |
| GA | 0.66 | 0.82 | 0.02 | 0.23 |
| GD | 0.77 | 0.94 | 0.02 | 0.35 |
| UKL | 0.52 | 0.60 | 0.02 | 0.24 |
| DPO | 0.63 | 0.98 | 0.05 | 0.90 |
| NPO | 0.65 | 0.84 | 0.04 | 0.89 |
| LUNAR | **0.25** | **0.97** | **0.04** | **0.95** |

## G  Limitations and Future Work

`LUNAR` relies on base models that are aligned to exhibit the ability to acknowledge a lack of knowledge or, at minimum, express their inability to respond. While such capabilities are common among mainstream models, they may not be present in raw, unaligned models. Future work could explore reference datasets ($D_{\text{ref}}$) with improved effectiveness of activation redirection. This study also represents an initial step toward bridging recent advances in LLM interpretability with robust unlearning. Further research may investigate how other interpretability tools can enhance unlearning effectiveness and controllability, contributing to the development of more reliable and principled unlearning methodologies.

## H  Broader Social Impact

This paper is motivated by the social consequences of recent advances in the field of machine learning and large language models (LLMs). LLMs have made significant strides by pre-training on and memorizing vast amounts of textual data. Additionally, information can also be incorporated during at post-training stage. Together, these processes can raise privacy and safety concerns. Consequently, the ability to efficiently remove data as specific as certain knowledge instances (such as those related to individual users) from these models, without compromising their predictive quality, is becoming increasingly important. We aim to provide a better and more efficient method to tackle this problem and enhance privacy and safety considerations in this field. Overall, we believe the potential positive social benefits of our work in LLM unlearning outweigh the potential negatives, which stem primarily from misuse.

