# OpenReview forum: "LLM Unlearning via Neural Activation Redirection"
_NeurIPS.cc/2025/Conference — NeurIPS 2025 poster_

### Official Review · Reviewer_7D8U · 2025-07-01

**Clarity:** 3
**Significance:** 2
**Originality:** 1
**Rating:** 4
**Confidence:** 4

**Summary:**

The paper proposes LUNAR, a closed-form and parameter efficient approach for machine unlearning in large LMs. Instead of fine-tuning the entire network or relying on contrastive pairs, the method computes a single unlearning vector at a carefully chosen layer, then optimizes only one MLP down-projection matrix so that activations for the forget set are steered into regions that naturally trigger the model to say it lacks the answer (by choosing the unlearning vector to be, for instance, the mean activation over refusal instances). The method results in up to 11.7 × better deviation scores than prior gradient-ascent, DPO and RMU baselines on PISTOL and TOFU, 20 × training-time speed-ups, and strong controllability, where models answer retained questions normally while refusing orto forgotten ones without hallucinations. The authors show robustness over several models under paraphrase and quantization adversarial attacks. PCA visualization shows that the method induces a separation between the representations of the forget and and retain set.

**Questions:**

How does the PCA in Figure 2 look like for the original model, before unlearning? doesn't it also separates the two sets?

**Ethical Concerns:**

["NO or VERY MINOR ethics concerns only"]

**Final Justification:**

I previously had concerns regarding the novelty compared with the previous work. The authors' rebuttal answers these concerns, and I think the paper can be accepted, provided that the relation to previous work is explained in the camera ready version.

**Limitations:**

Yes

**Quality:**

3

**Strengths And Weaknesses:**

The work is technically sound, well-motivated, and delivers a simple and powerful practical tool that tackles the utility–forget trade-off in unlearning. My main concern is novelty. Particularly, it is not clear to me what the inherent differences between the presented approach and knowledge editing methods such as ROME [1]. Yes, ROME is usually used for replacing a fact with a different fact, while here the goal is to replace a fact with an answer that shows refusal; but is this the whole difference? One can think of a variant of ROME where the target value vector v* is taken to be the mean activation over refusal answers. ROME also has a closed-form solution for this objective, i.e., we can find a low-rank update to U such that k*W = v*, where k is the activation over a prompt that aims to trigger the erase language. The present paper does not seek a low-rank update but rather re-learns the projection matrix, but this does not seem like a very significant difference. Do we have a reason to expect the presented method would do better? At the very least, I would expect to see a comparison with this baseline. In that sense the work is pretty similar to [2] which also uses a variant of ROME for unlearning.

In lines 98-105, the objective of matching the target vector that is calculated over an arbitrary retain set is presented as a novel contribution: “we conjecture that it is not strictly necessary for two features to be explicitly contrastive in a human-comprehensible sense to compute and utilize ‘steering vectors’. Instead, those can be employed more generally to map a shared hidden feature underlying one group of prompts (i.e., the source feature abstracted by the transformer in intermediate layers) to another group of prompts (i.e., the target feature).” However, the idea of being able to map one set of representations to another as a steering method, particularly under the linear representation hypothesis, has been extensively studied before in [3][4].

[1] Meng, Kevin, et al. "Locating and editing factual associations in gpt." Advances in neural information processing systems 35 (2022): 17359-17372.

[2] Hong, Yihuai, et al. "Intrinsic evaluation of unlearning using parametric knowledge traces." arXiv preprint arXiv:2406.11614 (2024).

[3] Ravfogel, Shauli, et al. "Linear adversarial concept erasure." International Conference on Machine Learning. PMLR, 2022.

[4] Singh, Shashwat, et al. "Representation Surgery: Theory and Practice of Affine Steering." International Conference on Machine Learning. PMLR, 2024.

---

> ### Author Rebuttal · Authors · 2025-07-31
>
> We thank the reviewer for acknowledging the technical soundness and practical contributions of LUNAR.
>
> We appreciate the opportunity to clarify its novelty. While the representation mapping is a general idea exploited by different works, there are substantial technical differences in the details of LUNAR and the works that the reviewer pointed out. Notably, LUNAR **uniquely addresses precise instance-level unlearning challenges caused by strong knowledge entanglement, as opposed to unlearning broad concept (line 27, 207))**. Additionally, LUNAR delivers **minimal parameter updates**, provides **robust defenses** against white-box attacks that prior unlearning methods fail, and **improves controllability** in producing dynamic, contextually-aware model behavior, clearly distinguishing our contributions from prior work.
>
> Below we address each point in turn and outline the additional material we will include in the camera-ready to make the distinctions more evident.
>
> **Comparison to ROME [1]:**
>
> Although we build on insights from ROME regarding the role of the down-projection ($W_{out}$), fundamental differences exist from multiple perspectives:
> - **Input computation**: ROME generates inputs to $W_{out}$ by averaging 50 randomly generated token sequences preceding the subject. In contrast, LUNAR directly uses activations from prompting the forget samples once, significantly simplifying the process.
> - **Target specification**: ROME explicitly modifies probabilities of specific tokens (requiring fixed target outputs). However, LUNAR considers predefining output tokens both unnecessary and a weakness, as it fails to emulate base model’s dynamic, contextually-aware response to unseen data. We find that finding model's internal understanding of "ignorance" and redirecting residual activations to this region is not only efficient but also highly effective at separating groups of highly entangled instances.
> - **Batched unlearning limitation**: adapting ROME from editing to unlearning, it is known to struggle with batched edits [5], a limitation that we find can be avoided through LUNAR’s activation-based approach.
>
> **Comparison to Intrinsic Evaluation [2]:**
>
> LUNAR has several key differences with the work [2] that the reviewer kindly pointed out. First, [2] introduces an evaluation methodology rather than an unlearning approach. Second, [2] evaluates the performance of unlearning a broad concept rather than specific instance-level samples. These substantial differences determine the different scope of [2] compared to LUNAR, thus outlining LUNAR’s novelty. More details follow.
>
> Concept unlearning methods, such as those discussed in [2], are insufficient for instance-level unlearning, which LUNAR tackles, due to strong entanglement caused by overlapping in tokens, semantics, and format. Specifically, [2] aims to identify vectors—located before the down-projection matrix—that are associated with vocabularies commonly used to express a given concept. Prior work has shown that such vectors act as keys for retrieving information stored in the down-projection matrix. Hence, fewer such vectors—resulting in fewer concept-related key tokens—reduce the model’s ability to retrieve information related to that concept.
>
> However, instance-level data points, which are the focus of LUNAR, exhibit strong entanglement between the forget and retain sets (e.g., unlearning relationship between entities A/B while retaining information about the relationship between A/C or B/D). The critical insight is that removing the influence of entity A and B tokens (and broader concept-related tokens) with concept unlearning methods could be counterproductive for retain set performance for instance-level unlearning.
>
> **Comparison to Representation Steering [3,4]:**
>
> Previous works explored steering between clearly defined contrastive concepts. Indeed in [4], steering is from source concept of male to target concept of female for classification task and from the source concept of toxic to target concept of nontoxic for generative task. There is a lack of work exploring critical scenarios where there is no clear source concept to begin with, as in the case of instance-level unlearning. LUNAR bridges this gap, showing that instance-level samples, despite high token, semantic and format similarity, can be cleanly separated via simple activation steering without explicit contrastive concept pairs. Moreover, LUNAR advances prior representation steering literature by:
> - Integrating steering directly within the existing model architecture rather than adding external steering modules.
> - Conducting robustness analyses under strong white-box attack scenarios and proposing mitigation strategies, neither of which has been addressed in prior steering works.
> - Demonstrating the effectiveness of simple activation addition compared to more complex affine steering procedures [4] for instance-level unlearning.
>
> Lastly, we highlight differences with LACE [3]. LACE also seeks to erase concepts (e.g., gender) rather than specific data points. It casts the concept unlearning problem as a minimax game between a projection P and an adversarial classifier on that one source concept - a setup clearly unsuited for instance-level unlearning. Other notable differences include, among others, that LACE is not a steering based method, it does not optimize base model parameters, and has a persistent cost (one matrix–vector product per application) which steering doesn’t have.
>
> We will further emphasize these differences and expand our comparisons in the camera-ready version.
>
> **Response to [Q1]**
>
> In the original model before unlearning, forget and retain set activations are not separable (i.e. the red dots cluster together with the green ones). The purpose of Figure 2 is to highlight that LUNAR effectively disentangles the two sets, despite strong original knowledge entanglement that caused suboptimal performance in prior SOTA instance-level unlearning methods.
>
> [5] Li et al. “Editing as Unlearning: Are Knowledge Editing Methods Strong Baselines for Large Language Model Unlearning?”

---

> > ### Comment · Reviewer_7D8U · 2025-08-01
> > **Response**
> >
> > Thanks for your response!
> >
> > Regarding the comparison with ROME: I agree with the technical differences you list. But, are they meaningful? One can think of a variant of ROME that uses less examples for fitting the low-rank update, for example. And there are followup works that aim to solve batched update. Have you compared with ROME or its variants?
> >
> > Regarding steering: I agree these are different problems. My point was about the claim you are the first to do something, while it was actually done in a somewhat different context.

---

> > > ### Author Response · Authors · 2025-08-01
> > >
> > > Thank you for engaging with our response!
> > >
> > > **Re. Difference with ROME:**
> > >
> > > First, we respectfully clarify that the differences from ROME are actually beyond technical. We have different loss functions applied on completely different spaces (token vs. activation).
> > >
> > > As we point out, LUNAR is not built on token-space manipulation as ROME is. The purpose of unlearning is to revert to a state where the model has not seen specific data points. As base models are aligned to dynamically express its ignorance to unseen data within the context (see examples in Table 1), users should expect the unlearned model to behave the same dynamically. ROME is an editing method instead - it requires concrete target answers to edit in token space. Hence, a user who applies this to unlearning will need to pre-define different token expressions, considering the context, for each unlearning data instance, which is less natural, efficient and scalable than LUNAR.
> > >
> > > This fundamental difference also addresses other critical issues that ROME and its variants find challenging, such as knowledge entanglement (also a critical issue for existing unlearning methods) and batched update (even if customized target responses are prepared for each individual data instance) as we mentioned.
> > >
> > > Second, regarding your question on batched updates, [5] that we provided above presents the latest advances in batched updates. This new ROME variant can only achieve retain set ROUGE1 score of 0.667 (its best case), whereas LUNAR achieves a perfect retain set ROUGE1 of 1.0. Our experiments confirmed this significant gap.
> > >
> > > **Re. Steering Claims:**
> > >
> > > We appreciate the opportunity to clarify our claim. We certainly don’t claim we invented representation steering. We discussed related steering works in Section 7, which [3] kindly mentioned by the reviewer was also built upon.
> > >
> > > However,  we believe the differences we highlight are significant and make practical differences though.
> > > - We demonstrate its applicability on a different use case which, in contrast to prior works, lacks contrastive concept pairs.
> > > - The paper introduces the notion of controllability in unlearning, that sets it apart from previous methods and addresses their known side-effects resulting from unboundedness or rigidity.
> > > - We also introduce a new loss function alongside an improved strategy for selecting intervention layers.
> > > - We assess its robustness against strong black-box and white-box attacks and contribute to propagating stress-testing unlearning methods.
> > > - Clear outperformance vs. existing methods in unlearning effectiveness, knowledge disentanglement, efficiency, and robustness.
> > >
> > > If the reviewer believes any aspect remains unclear or insufficient, we warmly invite any specific elaboration to help us provide additional clarity or analysis.
> > >
> > > We hope you will agree that these are all differentiating contributions that would deserve to be shared with the community of researchers and practitioners.

---

> > > ### Author Response · Authors · 2025-08-08
> > >
> > > Dear reviewer,
> > >
> > > Thank you again for your constructive feedback.
> > >
> > > Following your suggestion, we ran additional experiments comparing LUNAR with ROME, its batched-update variants, and other model editing methods adapted for unlearning.
> > >
> > > - Table 1: When unlearning only 20 instances, ROME collapses model utility (retain set ROUGE1 ≈ 0), showing the challenge for it to disentangle forget and retain data instances. Other editing methods either fail to unlearn (high forget set ROUGE1) or also collapse utility. The results further reaffirm that LUNAR preserves model utility while achieving complete unlearning, even with reduced batch sizes for editing baselines.
> > >
> > > - Table 2: Against the latest ROME-variant [5] that improves batched update (merge queries n=5, reported as most effective), LUNAR maintains retain set ROUGE1 ≈ 1.0 when unlearning 80 factual instances, versus ≈ 0.7 for ROME and variants, showing significant gap in performance. The gap is even larger when unlearning 80 PISTOL instances, where knowledge entanglement is stronger due to higher format, semantic and token similarities.
> > >
> > > Beyond stronger unlearning performance, as discussed in the main paper and earlier rebuttal, LUNAR produces context-aware abstentions to unlearned data (e..g., *“I apologize, but I cannot provide information on the types of books written by Jaime Vasquez as I do not have access to his personal information or literary works.”*) as opposed to rigid, predefined output like *“I don’t know”*, allowing it to closely emulate base-model behavior on unseen data.
> > >
> > > We believe this new evidence, together with our earlier responses, directly addresses your concerns and further underscores LUNAR’s advantages. We would be grateful if you could consider increasing your score. We remain happy to provide further clarification, experiments, or analysis should you have any specific questions.
> > >
> > > Thank you again for your time and valuable guidance.
> > >
> > > ---
> > >
> > > **Table 1:**
> > >
> > > | **PISTOL** | Forget Set ROUGE1 | Retain Set ROUGE1 |
> > > |----------------------------------|-------------------|-------------------|
> > > | **LUNAR**                        | **0.007**         | **0.922**         |
> > > | ROME                             | 0.000             | 0.000             |
> > > | MEMIT                            | 0.000             | 0.000             |
> > > | GRACE                            | 0.995             | 1.000             |
> > > | WISE                             | 0.684             | 0.941             |
> > > | AlphaEdit                        | 0.050             | 0.250             |
> > >
> > > | **Factual dataset** | Forget Set ROUGE1 | Retain Set ROUGE1 |
> > > |--------------------------------------------|-------------------|-------------------|
> > > | **LUNAR**                                  | **0.000**         | **0.981**         |
> > > | ROME                                       | 0.008             | 0.040             |
> > > | MEMIT                                      | 0.015             | 0.011             |
> > > | GRACE                                      | 0.653             | 0.815             |
> > > | WISE                                       | 0.284             | 0.762             |
> > > | AlphaEdit                                  | 0.079             | 0.686             |
> > >
> > >
> > > **Table 2:**
> > >
> > > | **PISTOL**      | Forget Set ROUGE1 | Retain Set ROUGE1 |
> > > |-----------------|-------------------|-------------------|
> > > | **LUNAR**       | **0.006**         | **0.917**         |
> > > | ROME-variant    | 0.010             | 0.295             |
> > > | MEMIT-variant   | 0.014             | 0.340             |
> > >
> > > | **Factual dataset** | Forget Set ROUGE1 | Retain Set ROUGE1 |
> > > |---------------------|-------------------|-------------------|
> > > | **LUNAR**           | **0.000**         | **0.975**         |
> > > | ROME-variant        | 0.045             | 0.667             |
> > > | MEMIT-variant       | 0.054             | 0.700             |

---

> > > > ### Comment · Reviewer_7D8U · 2025-08-08
> > > > **Response**
> > > >
> > > > Sorry for the delayed response. This answers my concerns, and I raise my score to 4.

---

> > > > > ### Author Response · Authors · 2025-08-08
> > > > >
> > > > > We sincerely appreciate your thoughtful assessment of our response and the decision to raise your score.
> > > > >
> > > > > We are glad that our response has addressed your concerns. Your insights have been instrumental in strengthening the work, and we will incorporate the additional results and discussions into the revised manuscript. Thank you again for the care and attention you have given to our submission.

---

> ### Author Response · Authors · 2025-08-05
>
> Dear Reviewer,
>
> Thank you again for your constructive engagement with our response, which is instrumental in improving the clarity on our contributions and distinguishing our approach from related lines of research.
>
> We are writing to kindly ask whether our rebuttal has sufficiently addressed your concerns.
>
> **To summarize**:
>
> [RE our method vs editing]:
> We have elaborated on how unlearning instance-level knowledge requires the practical ability to dynamically emulate the base model’s behavior, since the unlearned model must behave as it had never encountered the forget data. We also explained the limitations of token-space manipulation in effectively disentangling forget and retain instances. These critical requirements set our work apart from prior research in model editing and contrastive concept steering, both in terms of training objectives, implementation, and practical impact on performance and robustness.
>
> [RE contributions]:
> We have also highlighted our contributions across several dimensions including, but not limited to, the introduction of “controllability” as a key objective in unlearning, applicability of steering in instance-level unlearning, novel loss functions and implementation strategies, and the resultant state-of-the-art performance compared to existing instance-level unlearning baselines from multiple evaluation perspectives.
>
> We sincerely hope our responses have addressed your concerns. If so, we would be very grateful if you could consider revisiting your score. We remain committed to provide further clarification, experiments, or analysis should you have any specific questions.
>
> Thank you once again for your time and valuable guidance!
>
> Warm regards,
>
> The Authors

---

### Official Review · Reviewer_u6fC · 2025-07-02

**Clarity:** 2
**Significance:** 2
**Originality:** 3
**Rating:** 4
**Confidence:** 4

**Summary:**

The paper introduces a formal definition of controllability for the unlearning task and proposes LUNAR, a novel method that achieves state-of-the-art effectiveness and controllability while remaining memory- and compute-efficient and robust against various attacks.

**Questions:**

1.	Can you illustrate the benefits of the controllability in practical applications of machine unlearning?

2.	Is Layer Selection (Section 3.2) performed completely before Unlearning via Neural Activation Redirection (Section 3.1)? It seems that Neural Activation Redirection depends on the selected layer. If so, I recommend moving Section 3.2 before Section 3.1

3.	Does LUNAR chooses only one layer in the layer selection step? Can you explain why not choose two or more layers but only one?

4.	What properties of the down-projection layer make it the best target for unlearning, and how do other layers perform?

5.	Why is the similarity score chosen over metrics like L₂-norm in the layer selection, and how sensitive are results to this choice? Can you give some experiments to illustrate these?

6.	More experiments including the metric ES would be helpful, because it is important as shown in recent work [1].

**Ethical Concerns:**

["NO or VERY MINOR ethics concerns only"]

**Final Justification:**

I increase the score from 3 to 4

The authors have addressed:

1. Experiments about ES metric

2. It also can intervene multiple layers

3. Explain why select down-projection matrix

Part of resolved: explain why controllability is critical for practical applications

**Limitations:**

The same as Weakness 2

**Quality:**

3

**Strengths And Weaknesses:**

Strengths

1.	Provides a full derivation and proof of the closed-form solution, giving strong theoretical grounding.

2.	Fills an important gap by quantifying controllability in unlearning,

3.	Demonstrates low memory and computational overhead, making LUNAR practical for resource-constrained environments.

Weaknesses

1.	The description of layer selection in Section 3.1 feels abrupt and can confuse readers.

2.	The practical importance of the controllability definition isn’t illustrated with real-world scenarios, weakening the motivational impact.

3.	The relationship and order between layer selection (Section 3.1) and neural activation redirection (Section 3.2) are not clearly explained.

4.	The motivation of some parts in the method (e.g., layers and cosine similarity) is not clear. Please see Questions below.

5.	Evaluation lacks the recently proposed ES metric for LLM unlearning.

---

> ### Author Rebuttal · Authors · 2025-07-31
>
> We thank the reviewer for your valuable feedback and for recognizing the strong theoretical grounding of LUNAR, its efficiency, and its contribution in addressing an important gap by quantifying controllability in unlearning. Below, we provide our detailed response, and hope it addresses your questions and concerns.
>
> **[W2/Q1]:**
>
> We agree that explicitly illustrating controlled unlearning’s practical benefits strengthens the motivation. Specifically, controllability is critical for practical applications in several ways:
>
> **1. Alignment with user expectations and regulatory initiatives:**
> Real-world AI systems are increasingly designed to express their lack of knowledge explicitly and coherently. This aligns not only with user expectations but also recent regulatory initiatives advocating reliable, transparent, and safe AI behavior. Without controllability, models may generate hallucinations or nonsensical outputs in response to forget queries, as exemplified clearly in Table 1 of the paper.
>
> For example, in the medical domain, if a patient’s record is unlearned from the system, we expect the model to explicitly acknowledge its lack of knowledge when queried, rather than hallucinate or fabricate the patient’s history. Generating uncontrolled responses in such high-stakes settings poses significant risks, potentially leading to highly harmful or misleading outcomes. This underscores the importance of unlearning methods like LUNAR that prioritize controllability.
>
> **2. Dynamic and context-aware responses:**
> While some existing methods (e.g., DPO) mitigate hallucinations by generating fixed, static “I don’t know” responses, LUNAR improves controllability by producing natural, dynamic, and contextually appropriate acknowledgments of ignorance, closely matching base model behavior. This ensures higher-quality interactions, greatly enhancing user experience when interacting with unlearned models in practice.
>
> We provided systematic and comprehensive discussions on the practical implications and limitations of uncontrolled unlearning in the Appendix (lines 678–743). We will bring a summary of this discussion into the main text to highlight these practical benefits of controllability in the camera-ready version.
>
> **[W1/W3/Q2]:**
>
> We thank the reviewer for the helpful suggestions. We agree that the current structure of Section 3.1 and 3.2 could benefit from clearer exposition regarding the sequential flow of the LUNAR pipeline. To clarify, LUNAR first computes an unlearning vector for each layer using Eq.5 in Section 3.1, based on a predefined forget set and a reference set. Then, we select the most effective intervention layer (based on the method detailed in Section 3.2). Finally, we optimize the down-projection matrix of the selected layer using the loss defined in Eq.7.
>
> Our intention in the original ordering was to provide an overarching view of the full unlearning mechanism in Section 3.1 before delving into details of layer selection in Section 3.2. However, we acknowledge that this ordering may feel abrupt without explicitly stating the dependency between these steps. In the camera-ready, we will clarify this sequence at the beginning of Section 3 and explicitly state that layer selection is performed after the unlearning vector is computed but before the redirection is optimized. We believe this reordering of explanation and improved signposting will alleviate the confusion raised.
>
> **[Q3]:**
>
> We clarify that LUNAR selects and intervenes on a single layer only if the model is deployed in a black-box manner and thus insulated from skip-layer attack - a choice for ultimate computational and memory efficiency.
>
> We considered the potential risk posed by white-box adversaries who might attempt to bypass intervention layer(s) (e.g., via layer-skipping attacks). In such scenarios, we defend against this threat by selecting and modifying multiple top-ranked layers (e.g., top-3 for Llama2-7B), as detailed in Table 4 and Section 6.4. In this case, even if the attacker skips all intervention layers, no unlearned information will be recovered. This strategy maintains robustness without significantly compromising efficiency.
>
> In summary, we recommend practitioners adopting LUNAR to intervene on multiple layers for robustness under white-box threat models, while single-layer intervention is sufficient and most efficient in the common black-box deployment scenario.
>
> **[Q4]:**
>
> LUNAR targets the down-projection matrix for two key reasons: (1) its central role in factual knowledge storage (line 123), and (2) its suitability for efficient and stable optimization (line 194-196).
>
> 1. Prior work [1] has shown that the down-projection matrix within MLPs encodes factual associations, making it a natural site for intervention. Intervening this matrix allows LUNAR to directly overwrite internal representations of forget data. In contrast, other components (e.g., up projection or attention) are more involved in routing “keys”. Blocking the transport of “keys” is not the aim of unlearning.
> 2. From an optimization perspective, intervening on the down-projection matrix reduces the problem to a linear setting with a convex and smooth loss function (see Appendix B.2). This enables stable and efficient training.
>
> **[W4/Q5]:**
>
> Thanks for the question. We choose cosine similarity over L₂ norm for layer selection because it better captures semantic alignment with our target refusal behaviors. Specifically, we use Llama models to calculate cosine similarity, which normalizes the last-layer representation before the unembedding matrix, so they are trained to encode (token-level) semantic information in the direction, irrespective of the norm of the vector. Cosine similarity measures the angle between sentence embeddings, emphasizing direction (semantic meaning), which is crucial for evaluating whether a model’s response meaningfully aligns with desired refusals. This choice aligns more directly with the goal of producing coherent and contextually appropriate refusals.
>
> **[W5/Q6]:**
>
> We have conducted additional experiments on the ES metrics as suggested by the reviewer. As shown in the Table below, on the PISTOL dataset, all baseline methods fail to reduce the forget set ES score below 0.5, whereas LUNAR achieves a substantially lower forget set ES score of 0.25 while preserving a high retain set ES score of 0.97. On the TOFU dataset, all methods, including LUNAR, effectively reduce the ES score on the forget set to near zero. However, baseline methods such as GA, GD, and UKL exhibit a significant drop in the retain set ES score to around 0.3, indicating poor preservation of retained knowledge. Although DPO and NPO demonstrate performance comparable to LUNAR in terms of ES scores, they lack the key advantage of controllability. As discussed in the main paper, LUNAR ensures that the unlearned model abstains from answering rather than hallucinating incorrect content—a distinction not achieved by the baselines.
>
> In summary, ES scores reaffirm that LUNAR consistently outperforms all baseline methods. We will include the results in the camera-ready.
>
> | Method | PISTOL Forget Set ES | PISTOL Retain Set ES | TOFU Forget Set ES | TOFU Retain Set ES |
> |--------|:--------------------:|:--------------------:|:------------------:|:------------------:|
> | GA     |        0.66          |        0.82          |        0.02        |        0.23        |
> | GD     |        0.77          |        0.94          |        0.02        |        0.35        |
> | UKL    |        0.52          |        0.60          |        0.02        |        0.24        |
> | DPO    |        0.63          |        0.98          |        0.05        |        0.90        |
> | NPO    |        0.65          |        0.84          |        0.04        |        0.89        |
> | LUNAR  |        0.25          |        0.97          |        0.04        |        0.95        |
>
> [1] Meng et al "Locating and editing factual associations in gpt"

---

> ### Author Response · Authors · 2025-08-05
>
> Dear Reviewer,
>
> Thank you again for your insightful feedback, which has been instrumental in improving the clarity and strength of the paper.
>
> We are writing to kindly ask if our rebuttal and new results have sufficiently addressed your concerns. In particular, we would greatly value your thoughts on the following updates:
>
> - New Evaluation Results: Our additional experiments on ES scores re-confirm LUNAR’s **state-of-the-art performance vs baseline methods**.
>
> - Benefits of “controllability” in unlearning: We elaborated on the motivation for introducing “controllability” as a critical objective for unlearning research and deployment, particularly through examples in high-stakes healthcare setting. We believe this requirement **addresses common shortcomings** of prior unlearning methods and **supports safer and more practical deployment**.
>
> - Clarification of Implementation Details: We clarified intervention strategy and other implementation details raised by the reviewer.
>
> We sincerely hope that our responses and new results have addressed your concerns. If so, we would be very grateful if you might consider revisiting your score. Of course, we are also happy to provide any further evidence, experiments, or analysis should you have additional questions or recommendations.
>
> Thank you once again for your time and valuable guidance!
>
> Warm regards,
>
> The Authors

---

> > ### Comment · Reviewer_u6fC · 2025-08-06
> >
> > Thank you for your reply. It has addressed most of my concerns. I will increase my score to 4

---

> ### Author Response · Authors · 2025-08-06
>
> Thank you for your positive assessment of our rebuttal and for raising your score.
>
> We are glad our response addressed your concerns, and we sincerely appreciate your insightful feedback, which has been invaluable in strengthening our work.

---

### Official Review · Reviewer_E1m9 · 2025-07-04

**Clarity:** 2
**Significance:** 2
**Originality:** 2
**Rating:** 4
**Confidence:** 4

**Summary:**

In this paper, authors conduct unlearning on the forget dataset by updating the weight of MLP block within a single layer so that the forget data moves closer to the refusal direction regions. Authors method is lightweight without complex or conflicting loss objectives and is memory efficient due to low number of trainable parameters. Authors validate their method on several benchamkrs and also study their method effecgivness in white-box defense aware attack setting.

**Questions:**

•	It is not clear which layer’s activations are used to plot Figure 2. Is this the intervention layer or the final layer?

•	The authors could provide activation patterns from the intervention layer through to the final layers, for both the retain and forget sets, contrasting the base and unlearned models. How do the activations for the retain dataset shift in layers following the intervention,   do they lie close apart ?

•	What is the impact if the retain dataset is used only for computing the steering vectors, and not for MLP training in the unlearning step? Moreover, what happens if the retain dataset is unavailable—can the steering method still function effectively?

•	In the reverse direction attack, it looks more appropriate to compute the steering vector on the unlearned model rather than reusing the reversing original steering vector from the base model. Can the authors recompute the steering vector between retain and forget sets using the unlearned model and then add that vector to push the forget data closer to the retain set?

•	The evaluation primarily focuses on the forget and retain datasets. However, when redirecting the forget data toward refusal regions, how do we ensure that the model's utility is not harmed on unrelated datasets? Can the authors report model performance on diverse benchmarks (e.g., ARC, PiQA, SciQ, OpenBookQA) before and after unlearning? Please clarify if this is already covered within the unlearning benchmarks.

•	In the layer-skip attack, the authors skip the layer that is intervened. As discussed in the paper, prior work shows that skipping layers [6, 12, 13] results in minimal performance drop. It is unclear why unlearning remains effective on the forget dataset in this setting. Although the authors mention a "top-3 intervention," the evaluation setup lacks sufficient explanation. Please discuss the setting with more details.

• How do unlearned model performs attacks such as white-box Logitlens attack like in [a]?


[a] Can Sensitive Information Be Deleted from LLMs? ICLR 2024

**Ethical Concerns:**

["NO or VERY MINOR ethics concerns only"]

**Final Justification:**

For the reasons mentioned in my rebuttal comment, I increase the score to BA.

**Quality:**

2

**Strengths And Weaknesses:**

**Strengths**

•	The method is easy to implement and uses a single loss function, unlike previous works that employ separate loss formulations for the forget and retain datasets. Moreover, the training is lightweight, as it only optimizes a single MLP down-projection layer within each transformer block.

•	The authors validate their unlearning defense against adaptive white-box attacks to showcase the robustness of the proposed method.

•	The paper offers valuable insights into how to conduct precise interventions to enable forgetting, which is a fundamental direction with  a strong potential for future work to build upon.

**Weaknesses**

•	While the method is highly efficient due to its precise intervention, it modifies only a single layer, leaving the rest of the model unchanged. I have concerns about whether an attacker could still exploit the untouched layers to extract information about the forget dataset.

•	The paper lacks deeper analysis of how the initial and unlearned models differ across layers. Specifically, the behavioral shifts,how activations propagate or change beyond the intervention layer are not thoroughly discussed.

---

> ### Author Rebuttal · Authors · 2025-07-31
>
> We thank the reviewer for their valuable feedback and for recognizing that our work offers important insights into conducting precise interventions for instance-level unlearning while maintaining efficiency and robustness against white-box attacks. We sincerely hope our responses below clarify your concerns and questions. Please let us know if there is anything further we can address.
>
> **[W1]:**
>
> Thank you for highlighting this important aspect. We considered this risk, and experimented with a variant that intervenes on the top-K layers identified in the layer selection process (we set K = 3, see Section 6.4). We validated the robustness of this variant to a set of attacks that include a ‘layer-skip’ attack, among others (LogitLens, Quantization, Paraphrasing, Reverse-direction). The results, in Table 4, demonstrate that this targeted intervention robustly defends against SOTA attacks that exploit untouched layers. We will better highlight this aspect in the paper.
>
> Further, following your suggestion, we conducted additional experiments using the white-box logit lens attack, which further corroborated LUNAR’s resilience (See details in response to Q7 below).
>
> LUNAR’s deliberate design to intervene in early-to-middle layers, where initial information accumulation and abstract feature formation occur [1], prevents meaningful internal representations of the forget set from forming, thereby achieving robustness alongside efficiency.
>
> **[Q1]:**
>
> Sorry for the confusion. The plots correspond to the activation space immediately post intervention. Forget and retain activations remain separated through to the last layer.
>
> **[W2/Q2]:**
>
> Thank you for pointing us to this valuable analysis. In response to your suggestion, we analyzed the layer-wise average L2 distance of data point activations between models before and after LUNAR unlearning for both forget and retain set through all layers.
>
> Our results show a clear step wise increase in the average L2 distance for the forget set immediately after the intervention layer. For example in Llama2-7B, it rises from 0 to approximately 0.01 at the intervention layer 18 and continues to increase, exceeding 0.05 by the final layer. In contrast, the average L2 distance for the retain set remains very close to 0 immediately after the intervention and stayed almost unchanged through to the final layer. This behavior is consistent across base models. It indicates that LUNAR’s intervention is highly localized (well separate the forget and retain set despite knowledge entanglement) and the retain set activations stay close through to the final layer.
> | Layer | Retain Set | Forget Set |
> |-------|------------|-------------|
> | 16    | 0.000      | 0.000       |
> | 17    | 0.000      | 0.000       |
> | 18    | 0.000      | 0.010       |
> | 19    | 0.000      | 0.011       |
> | ... | ...        | ...         |
> | 24    | 0.000      | 0.017       |
> | 25    | 0.000      | 0.020       |
> | 26    | 0.000      | 0.024       |
> | ... | ...        | ...         |
> | 29    | 0.000      | 0.036       |
> | 30    | 0.001      | 0.042       |
> | 31    | 0.001      | 0.052       |
> We will include these findings and visualizations in the camera-ready version.
>
> **[Q3]:**
>
> We would like to clarify two points:
>
> First, as shown in Eq.5, the retain set is not utilized when computing the unlearning vector; only the “center of mass” of forget-set activations determines the starting point for activation redirection.
>
> Second, we include only as many randomly selected retain samples as there are in the forget set during optimization (Eq. 6), primarily to prevent optimization from drifting retain set activations. Following your suggestion, we conducted further ablation experiments. Completely omitting retain samples in Eq. 6 would cause unintended over-refusal on the retain set caused by the drift of retain set activations - refusal score of the retain set is 0.548 (unlearn PISTOL with Llama-7B as the base model). This supports the more robust performance of LUNAR in its current setup.
>
> We believe that including only minimal retain samples is a practical, realistic scenario. We are happy to add this ablation to the paper if the reviewer finds it useful.
>
> **[Q4]:**
>
> Thanks for this interesting suggestion. We have performed the attack as you described. For Llama-7B, this attack resulted in a ROUGE1 score of only 0.025 on the forget set (similar for other base models), confirming no information being recovered and LUNAR’s robustness against this attack variant. We appreciate this valuable input and will incorporate relevant analyses and discussions into the camera-ready.
>
> **[Q5]:**
>
> Thanks for this suggestion. We have conducted the analysis - results show that performance remains at the same level of the base mode, confirming that LUNAR performs targeted, minimally invasive interventions. We include results on reviewer’s suggested metrics below and will include comprehensive results across model families in the updated version.
>
> |               | ARC-Easy | ARC-Challenge | PiQA  | SciQ  | OpenBookQA |
> |---------------|:--------:|:-------------:|:-----:|:-----:|:----------:|
> | Llama2-7B-chat |  0.717   |     0.462     | 0.773 | 0.898 |   0.438    |
> | LUNAR         |  0.724   |     0.470     | 0.763 | 0.903 |   0.432    |
>
> **[Q6]:**
>
> Thank you for highlighting this point, which helps clarify an important aspect of our evaluation setup.
>
> In the standard deployment scenario where skip-layer attacks are not a threat (e.g., black-box access), LUNAR intervenes on a single layer to maximize efficiency.
>
> However, if the risk of skip-layer attack is real, we modify multiple layers using the same methodology. As shown in Table 4, modifying the top-3 layers (identified using the same selection criteria described in Section 3.2) is sufficient to defend against skip-layer attacks. That is even if all modified layers are skipped, the model does not reveal information from the forget set. Importantly, multi-layer intervention maintains the same level of unlearning performance and controllability, though it introduces additional computational cost.
>
> We appreciate your suggestion and will clarify this in more detail, emphasizing that practitioners should adapt the number of intervened layers based on the anticipated risk of skip-layer attack. We will include expanded discussions to clearly differentiate these scenarios in the camera-ready version.
>
> **[Q7]:**
>
> Thanks for the suggestion. In the paper, we already evaluate LUNAR against four SOTA attacks, demonstrating strong robustness to recovery of the forget set. To further strengthen this analysis, we additionally applied LogitLens to assess whether the model still retains information about the forgotten content.
>
> We conducted this analysis across several layers and reported representative results for layer 17 (before intervention), layer 18 (after intervention), and layer 32 (final layer). At layers 17 and 18, LogitLens produces only unrelated or gibberish tokens, indicating that forget set information is not recoverable immediately before or after the intervention point. At the final layer, the top prediction is the token “I”, consistent with LUNAR’s intended redirection toward refusals (e.g., “I apologize…”). These results further confirm that LUNAR effectively redirects memory traces of the forget set even under direct activation-to-logit mapping.
>
> **Prompt:** *What was the effective date of the contract between Wnzatj SAS and Jzrcws SA?*
> **Expected Answer:** `06-02-1998`
>
> | Layer     | Rank | Token     | Probability |
> |-----------|------|-----------|-------------|
> | Layer 17  | 1    | '▸'       | 0.042969    |
> |           | 2    | 'uf'      | 0.010498    |
> |           | 3    | 'address' | 0.005463    |
> |           | 4    | 'Collins' | 0.003525    |
> |           | 5    | 'ribu'    | 0.003418    |
> | Layer 18  | 1    | 'Ans'     | 0.011536    |
> |           | 2    | 'answer'  | 0.008728    |
> |           | 3    | '▸'       | 0.006775    |
> |           | 4    | 'ribu'    | 0.005127    |
> |           | 5    | 'Unis'    | 0.003738    |
> | Layer 32  | 1    | 'I'       | 0.1602      |
> |           | 2    | 'eth'     | 0.0168      |
> |           | 3    | 'Eth'     | 0.0140      |
> |           | 4    | 'quelle'  | 0.0085      |
> |           | 5    | 'dd'      | 0.0082      |
>
> [1] Gross et. al. “Studying large language model generalization with influence functions”

---

> ### Author Response · Authors · 2025-08-05
>
> Dear Reviewer,
>
> Thank you again for your insightful feedback, which has been instrumental in strengthening the paper.
>
> We are writing to kindly ask if our rebuttal and new results have sufficiently addressed your concerns. In particular, we would greatly appreciate your thoughts on the following updates:
>
> - Clarification on Intervention Strategy: We clarified the intervention setup, highlighting that modifying the top-K layers (K=3) offers **robust protection** against white-box attacks.
>
> - New Robustness Results: Our additional experiments on logic-lens and variant reverse-direction attacks re-confirm LUNAR’s **strong robustness**.
>
> - Deeper Activation Dynamics Analysis: Our deeper analysis of activation evolution across layers confirms that forget and retain instances remain **well disentangled**, providing further evidence of the **robustness and precision of our intervention**.
>
> - Downstream Task Evaluation: New experiments demonstrate that LUNAR’s interventions are highly localized with **no impact on the model’s general performance**, reinforcing the practicality of our approach.
>
> We sincerely hope that our responses and new results have addressed your concerns. If so, we would be very grateful if you might consider revisiting your score. Of course, we are also happy to provide any further evidence, experiments, or analysis should you have additional questions or recommendations.
>
> Thank you once again for your time and valuable guidance!
>
> Warm regards,
>
> The Authors

---

> > ### Comment · Reviewer_E1m9 · 2025-08-08
> >
> > Sorry for my late response.
> >
> > I went through all your rebuttal comments in detail. My concerns are now fairly addressed, with empirical evidence provided in [W2/Q2] showing a shift in activations only for the forget set, and in [Q5] demonstrating utility on other benchmarks, and more experiments with logit-lens attack [Q7].
> >
> > I suggest the authors incorporate all the additional experiments into the revised paper, including the ablation on the retain set in Eq. [6]. Please also try to provide more visualizations contrasting the base and unlearned models, in addition to the distances reported in [W2/Q2].
> >
> > Finally, I am increasing my score to BA, as all my comments have been addressed. I hope this work serves as a strong benchmark for defending against novel unlearning attacks in the future. Please make the code available for the benefit of the community.

---

> > > ### Author Response · Authors · 2025-08-08
> > >
> > > Thank you for your positive assessment of our rebuttal and for raising your score.
> > >
> > > We are pleased that our response has addressed all of your concerns. We agree and greatly appreciate your recognition that LUNAR could serve as a strong benchmark for defending against future, novel unlearning attacks. In addition, we believe that, given its strong effectiveness, LUNAR can also serve as a solid performance baseline for instance-level unlearning more broadly.
> > >
> > > We will follow your suggestion and incorporate all additional experimental results, ablation studies, and visualizations into the camera-ready version. Our code will also be publicly available to support adoption and benefit the community.
> > >
> > > Your insightful feedback has been invaluable in strengthening our work, and we are sincerely grateful for the time and care you have devoted to our submission.

---

### Note · Authors · 2025-08-11

We thank the reviewers for their thoughtful feedback and constructive engagement. We are glad that the discussion, supported by various new experiments, has addressed reviewer concerns. We are pleased that no outstanding issues remain, and that the positive score changes suggest consensus on the paper’s contributions, technical soundness, and relevance.

In summary, reviewers identified the following as key strengths of the paper:

- **Efficiency and Practicality:** LUNAR is efficient, easy to implement, and a simple and powerful practical tool for addressing the utility–forget trade-off in unlearning (E1m9, u6fC, 7D8U).
- **Robustness:** Robustness against adaptive white-box attacks, positioning LUNAR as a strong benchmark for defending against future, novel unlearning attacks (E1m9).
- **Contributions:** Valuable insights into precise interventions for enabling forgetting (E1m9), filling an important gap by quantifying controllability in unlearning (u6fC), and offering theoretical grounding and technical soundness (u6fC, 7D8U).

Our discussion and additional experiments directly addressed all reviewer concerns, most notably:

- **Controllability:** We illustrate its importance for instance-level unlearning with practical high-stakes application example (u6fC).
- **Robustness:** We clarified LUNAR’s top-K layer intervention strategy as an effective defense against white-box attacks, with added evaluations under logit-lens and reverse-direction variant attacks, reaffirming its robustness (E1m9, u6fC).
- **Locality:** We confirmed that LUNAR performs targeted, minimally invasive interventions via (1) deeper activation-evolution analysis of the initial and unlearned model (LUNAR cleanly separates highly entangled forget and retain data instances and preserves retain activations through the final layer), (2) significant ES-score improvements vs baselines (confirm stronger unlearning performance and robustness), and (3) downstream task evaluations (show no utility degradation) (E1m9, u6fC).
- **Clarified Scope and Differentiation:** We clarified and extended novelties and contributions beyond ROME, model editing variants, and prior steering methods, supported by significant performance gains (7D8U).

We are grateful for the reviewers’ time and suggestions, which we will incorporate, along with the new experimental results, into the revised version, further strengthening its contribution to the research and adoption of robust instance-level unlearning.

---

### Decision · Program_Chairs · 2025-09-17

**Decision:**

Accept (poster)

**Comment:**

This paper proposes a new unlearning methods. Reviewers were initially hesitant about the efficacy of the method and its practicality for language models, but the author rebuttals sufficiently addressed the reviewer concerns. Overall, the paper is at the very least a strong baseline for future unlearning methods - I strongly urge the authors to incorporate the reviewers' comments and the additional experiments from the rebuttals into the final paper.